# GSLB: The Graph Structure Learning Benchmark

**Zhixun Li**[1]   **Liang Wang**[2,3]   **Xin Sun**[4]   **Yifan Luo**[5]   **Yanqiao Zhu**[6]   **Dingshuo Chen**[2,3]
**Yingtao Luo**[7]   **Xiangxin Zhou**[2,3]   **Qiang Liu**[2,3*]   **Shu Wu**[2,3]   **Liang Wang**[2,3,4]   **Jeffrey Xu Yu**[1*]

[1] Department of Systems Engineering and Engineering Management
The Chinese University of Hong Kong
[2]Center for Research on Intelligent Perception and Computing
State Key Laboratory of Multimodal Artificial Intelligence Systems
Institute of Automation, Chinese Academy of Sciences
[3] School of Artificial Intelligence, University of Chinese Academy of Sciences
[4]Department of Automation, University of Science and Technology of China
[5]School of Cyberspace Security, Beijing University of Posts and Telecommunications
[6]Department of Computer Science, University of California, Los Angeles
[7]Heinz College of Information Systems and Public Policy, Machine Learning Department
School of Computer Science, Carnegie Mellon University
✉ Primary contact: `zxli@se.cuhk.edu.hk`

## Abstract

Graph Structure Learning (GSL) has recently garnered considerable attention due to its ability to optimize both the parameters of Graph Neural Networks (GNNs) and the computation graph structure simultaneously. Despite the proliferation of GSL methods developed in recent years, there is no standard experimental setting or fair comparison for performance evaluation, which creates a great obstacle to understanding the progress in this field. To fill this gap, we systematically analyze the performance of GSL in different scenarios and develop a comprehensive Graph Structure Learning Benchmark (GSLB) curated from 20 diverse graph datasets and 16 distinct GSL algorithms. Specifically, GSLB systematically investigates the characteristics of GSL in terms of three dimensions: **effectiveness**, **robustness**, and **complexity**. We comprehensively evaluate state-of-the-art GSL algorithms in node- and graph-level tasks, and analyze their performance in robust learning and model complexity. Further, to facilitate reproducible research, we have developed an easy-to-use library for training, evaluating, and visualizing different GSL methods. Empirical results of our extensive experiments demonstrate the ability of GSL and reveal its potential benefits on various downstream tasks, offering insights and opportunities for future research. The code of GSLB is available at: `https://github.com/GSL-Benchmark/GSLB`.

## 1   Introduction

Graphs, structures made of vertices and edges, are ubiquitous in real-world applications. A wide variety of applications spanning social network [51, 9], molecular property prediction [40, 14], fake news detection [45, 1], and fraud detection [23, 27] have found graphs instrumental in modeling complex systems. In recent years, Graph Neural Networks (GNNs) have attracted increasing attention due to their powerful ability to learn node or graph representations. However, most of the GNNs heavily rely on the assumption that the initial structure of the graph is trustworthy enough to serve as ground-truth for training. Due to uncertainty and complexity in data collection, graph structures are inevitably redundant, biased, noisy, incomplete, or the original graph structures are even unavailable, which will bring great challenges for the deployment of GNNs in real-world applications.

---

*Corresponding authors: Qiang Liu (qiang.liu@nlpr.ia.ac.cn), Jeffrey Xu Yu (yu@se.cuhk.edu.hk)

37th Conference on Neural Information Processing Systems (NeurIPS 2023) Track on Datasets and Benchmarks.

Table 1: An overview of GSLB. Both algorithms and datasets are divided into three categories: homogeneous node-level, heterogeneous node-level, and graph-level. The evaluation is divided into three dimensions: **effectiveness**, **robustness**, and **complexity**.

| | |
|---|---|
| ***Algorithms*** | |
| Homogeneous GSL | LDS [11], GRCN [48], ProGNN [18], IDGL [4], CoGSL [26], SUBLIME [28], GEN [39], STABLE [21], NodeFormer [43], SLAPS [10], GSR [54], HES-GSL [42] |
| Heterogeneous GSL | GTN [49], HGSL [53] |
| Graph-level GSL | HGP-SL [52], VIB-GSL [36] |
| ***Datasets*** | |
| Homogeneous datasets | Cora [47], Citeseer [47], Pubmed [47], ogbn-arxiv [15], Polblogs, Cornell [34], Texas [34], Wisconsin [34], Actor [37] |
| Heterogeneous datasets | ACM [49], DBLP [49], Yelp [29] |
| Graph-level datasets | IMDB-B [3], IMDB-M [3], COLLAB [46], REDDIT-B [46], MUTAG [5], PROTEINS [2], Peptides-Func [8], Peptides-Struct [8] |
| ***Evaluations*** | |
| Effectiveness | Homogeneous node classification (Topology Refinement/Topology Inference), Heterogeneous node classification, Graph-level tasks |
| Robustness | Supervision signal robustness, Structure robustness, Feature robustness |
| Complexity | Time complexity, Space complexity |

To mitigate the aforementioned problems, Graph Structure Learning (GSL) [4, 55, 30, 10, 57, 50] has become an important theme in graph learning. GSL aims to make the computation structure of GNNs more suitable for downstream tasks and improve the quality of the learned representations. While it is widespread in different communities and the research enthusiasm for GSL is increasing, there is no standardized benchmark that could offer a fair and consistent comparison of different GSL algorithms. Moreover, due to the complexity and diversity of graph datasets, the experimental setups in existing work are not consistent, such as varying ratios of the training set and different train/validation/test splits. This poses a great obstacle to a holistic understanding of the current research status. Therefore, the development of a standardized and comprehensive benchmark for GSL is an urgent need within the community.

In this work, we propose Graph Structure Learning Benchmark (GSLB), which serves as the first comprehensive benchmark for GSL. Our benchmark encompasses 16 state-of-the-art GSL algorithms and 20 diverse graph datasets covering homogeneous node-level, heterogeneous node-level, and graph-level tasks. We systematically investigate the characteristics of GSL in terms of three dimensions: **effectiveness**, **robustness**, and **complexity**. Based on these three dimensions, we conduct an extensive comparative study of existing GSL algorithms in different scenarios. For **effectiveness**, GSLB provides a fair and comprehensive comparison of existing algorithms on homogeneous node-level, heterogeneous node-level, and graph-level tasks, where we consider both homophilic and heterophilic graph datasets for homogeneous node-level tasks, and cover both Topology Refinement (TR, i.e., refining graphs from data with the original topology) and Topology Inference (TI, i.e., inferring graphs from data without initial topology) settings. For **robustness**, GSLB evaluates GSL models under three types of noise: supervision signal noise, structure noise, and feature noise. We also compare GSL algorithms with the models specifically designed to improve these types of robustness. For **complexity**, GSLB conducts a detailed evaluation of representative GSL algorithms in terms of time complexity and space complexity.

Through extensive experiments, we observe that: (1) GSL generally brings performance improvement for node-level tasks, especially on heterophilic graphs; (2) on graph-level tasks, current GSL models bring limited improvement and their performance varies greatly across different datasets; (3) most GSL algorithms (especially unsupervised GSL algorithms) show impressive robustness; (4) GSL models require significant time and memory overhead, making them challenging to deploy on large-scale graphs. In summary, we make the following three contributions:

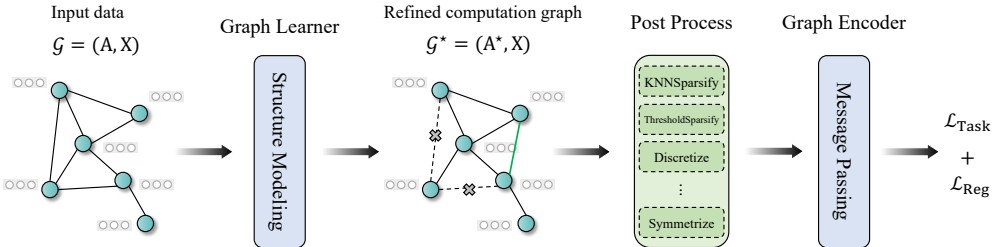

Figure 1: A general framework of Graph Structure Learning (GSL). GSL methods start with input features and an optional initial graph structure. Its corresponding computation graph is refined/inferred through a structure learning module. With the learned computation graph, Graph Neural Networks (GNNs) are used to generate graph representations.

- We propose GSLB, the first comprehensive benchmark for graph structure learning. We integrate 16 state-of-the-art GSL algorithms and 20 diverse graph datasets covering homogeneous node-level, heterogeneous node-level, and graph-level tasks. An overview of our benchmark is shown in Table 1.

- To explore the ability and limitations of GSL, we systematically evaluate existing algorithms from three dimensions: **effectiveness**, **robustness**, and **complexity**. Based on the results, we reveal the potential benefits and drawbacks of GSL to assist future research efforts.

- To facilitate future work and help researchers quickly use the latest models, we develop an easy-to-use open-source library. Besides, users can evaluate their own models or datasets with less effort. Our code is available at `https://github.com/GSL-Benchmark/GSLB`.

## 2 Problem Definition

In this section, we will briefly review the advances and basic concepts of GSL. Given an undirected graph $\mathcal{G} = (\mathbf{A}, \mathbf{X})$, where $\mathbf{A} \in \mathbb{R}^{N \times N}$ is the adjacency matrix, $a_{uv} = 1$ if edge $(u, v)$ exists and $a_{uv} = 0$ otherwise, and $\mathbf{X} \in \mathbb{R}^{N \times F}$ is the node features matrix, $N$ is the number of nodes, $F$ is the dimension of node features. Given an optional graph $\mathcal{G}$, the goal of GSL is to jointly optimize computation graph $\mathcal{G}^\star = (\mathbf{A}^\star, \mathbf{X})$ and the parameters of graph encoder $\Theta_f$ to obtain high-quality node representations $\mathbf{Z}^\star \in \mathbb{R}^{N \times F'}$ for downstream tasks, where $\mathbf{A}^\star$ is the refined graph by graph learner.

In general, the objective of GSL can be summarized as the following formula:

$$\mathcal{L}_{\text{GSL}} = \mathcal{L}_{\text{Task}}(\mathbf{Z}^\star, \mathbf{Y}) + \lambda \mathcal{L}_{\text{Reg}}(\mathbf{A}^\star, \mathbf{Z}^\star, \mathcal{G}) \tag{1}$$

where the first term $\mathcal{L}_{\text{Task}}$ refers to a task-specific objective with respect to the learned representation $\mathbf{Z}^\star$ and ground-truth $\mathbf{Y}$, the second term $\mathcal{L}_{\text{Reg}}$ imposes constraints on the learned graph structure and representations, and $\lambda$ is a hyper-parameter that controls the trade-off between the two terms. The general framework of GSL is shown in Figure 1.

## 3 GSLB: Graph Structure Learning Benchmark

In this section, we introduce the overview of Graph Structure Learning Benchmark, with considerations of algorithms (Section 3.1), datasets (Section 3.2) and evaluations (Section 3.3).

### 3.1 Benchmark Algorithms

Table 1 shows the overall 16 algorithms integrated in GSLB. They are divided into three categories: homogeneous GSL, heterogeneous GSL, and graph-level GSL. We briefly introduce each category in the following, and more details are provided in Appendix A.2.

**Homogeneous GSL.** Most of the existing GSL algorithms are designed for homogeneous graphs. They assume there is only one type of nodes and edges in the graph. We select 7 TR-oriented

algorithms including GRCN [48], ProGNN [18], IDGL [4], GEN [39], CoGSL [26], STABLE [21], and GSR [54]. For TI-oriented algorithms, we select SUBLIME [28], NodeFormer [43], SLAPS [10], and HES-GSL [42]. It is worth noting that TR-oriented algorithms can only be applied if the original graph structure is available, but we can construct a preliminary graph based on node features (e.g., $k$NN graphs or $\epsilon$-graphs).

**Heterogeneous GSL.** We integrate two representative heterogeneous GSL algorithms: Graph Transformer Networks (GTN) [49] and Heterogeneous Graph Structure Learning (HGSL) [53], which can handle the heterogeneity and capture complex interactions in heterogeneous graphs.

**Graph-level GSL.** Graph-level GSL algorithms aim to refine each graph structure in datasets. We select two graph-level algorithms: Hierarchical Graph Pooling with Structure Learning (HGP-SL) [52] and Variational Information Bottleneck guided Graph Structure Learning (VIB-GSL) [36].

### 3.2 Benchmark Datasets

To comprehensively and effectively evaluate the characteristics of GSL in the field of graph learning, we have integrated a large number of datasets from various domains for different types of tasks. For node-level tasks, to evaluate the most mainstream task of GSL, node classification, we use four citation networks (i.e., Cora, Citeseer, Pubmed [47]), and ogbn-arxiv [15], three website networks from WebKB (i.e., Cornell, Texas, and Wisconsin [34]), and a cooccurrence network Actor with homophily ratio ranging from strong homophily to strong heterophily. Subsequently, to validate the effectiveness of GSL in heterogeneous node classification, we utilized three heterogeneous graph datasets (i.e., DBLP [49], ACM [49], and Yelp [29]). To investigate the robustness of GSL, we further incorporate the Polblogs dataset for evaluation. For graph-level tasks, we select six public graph classification benchmark dataset from TUDataset [31] for evaluation, including IMDB-B [3], IMDB-M [3], RDT-B [46], COLLAB [46], MUTAG [5] and PROTEINS [2]. Each dataset is a collection of graphs where each graph is associated with a level. Besides, exploring whether GSL can capture long-range information is an exciting topic. Therefore, we have utilized recently proposed long-range graph datasets: Peptides-func and Peptides-struct [8]. See more details and statistics about datasets in Appendix A.1.

### 3.3 Benchmark Evaluations

To comprehensively investigate the pros and cons of GSL, our benchmark evaluations encompass three dimensions: **effectiveness**, **robustness**, and **complexity**. For **effectiveness**, GSLB provides a fair and comprehensive comparison of existing algorithms from three perspectives: homogeneous node classification, heterogeneous node classification, and graph-level tasks. In the case of homogeneous node classification, we evaluated them on both homophilic and heterophilic graph datasets, conducting experiments in both TR and TI scenarios. For graph-level, we evaluate graph-level GSL algorithms on TUDataset and long-range graph datasets for exploring the capabilities on graph-level tasks. For most datasets, we use accuracy as our evaluation metric. For **robustness**, GSLB evaluates three types of robustness: supervision signal robustness, structure robustness, and feature robustness. We control the count of labels to explore the supervision signal robustness of GSL and find that GSL exhibits excellent performance in the scenarios with few labels. We inject random structure noise and graph topology attacks to investigate the structure robustness. We also study the feature robustness by randomly masking a certain proportion of node features. For **complexity**, we conduct a detailed evaluation of representative GSL algorithms in terms of time complexity and space complexity. It will help to facilitate the deployment of GSL in real-world applications.

## 4 Experiments and Analysis

In this section, we systematically investigate the **effectiveness**, **robustness**, and **complexity** of GSL algorithms by answering the following specific questions:

- For **effectiveness**, **RQ1**: How effective are the algorithms on node-level representation learning (Section 4.2)? **RQ2**: Can GSL mitigate homophily inductive bias of traditional message-passing based GNNs (Section 4.2)? **RQ3**: How does GSL perform on heterogeneous graph datasets (Section 4.3)? **RQ4**: How effective are the algorithms on graph-level representation learning (Section 4.4)? **RQ5**: Can GSL methods capture long-range information on the graph (Appendix B)?

- For **robustness**, **RQ6**: How robust are GSL algorithms when faced with a scarcity of labeled samples? **RQ7**: How robust are GSL algorithms in the face of structure attack or noise? **RQ8**: How is the feature robustness of GSL? (Section 4.5)

- For **complexity**, **RQ9**: How efficient are these algorithms in terms of time and space (Section 4.6)?

- Otherwise, **RQ10**: What does the learned graph structure look like (Appendix B.2)?

## 4.1 Experimental Settings

All algorithms in GSLB are implemented by PyTorch [33], and unless specifically indicated, the encoders for all algorithms are Graph Convolutional Networks. All experiments are conducted on a Linux server with GPU (NVIDIA GeForce 3090 and NVIDIA A100) and CPU (AMD EPYC 7763), using PyTorch 1.13.0, DGL 1.1.0 [38] and Python 3.9.16.

## 4.2 Performance on node-level representation learning

For node-level representation learning, we conduct experiments on homogeneous graph datasets under both TR and TI scenarios, and use classification accuracy as our evaluation metric. Table 2 shows the experimental results of various GSL algorithms under the standard setting of transductive node classification task in the TR scenario. We can observe that: 1) Most GSL algorithms generally show improvements in node classification task, particularly on datasets with high heterophily ratio. Due to the presence of heterophilic connections in heterophily graphs, where nodes are often connected to nodes with different labels, it violates the homophily assumption of message-passing neural networks. As a result, traditional GNNs like GCN and GAT exhibit poor performance. However, GSL can improve significantly on heterophily graph datasets by learning new graph structures based on downstream tasks and specific learning objectives, thus enhancing the homophily of the graph and promoting the performance on node-level representation learning. 2) SUBLIME achieves optimal or near-optimal results on most datasets. It learns graph structure through contrastive learning in an unsupervised manner. As mentioned in the recent literature [10], optimizing graph structures solely based on label information is insufficient. Leveraging a large and abundant amount of unlabeled information can enhance the performance of GSL. 3) The scalability of GSL still needs improvement, as only a few models can be trained on large-scale datasets (e.g., ogbn-arxiv). We will discuss the scalability of GSL algorithms in detail in a subsequent section (Section 4.6).

Table 3 shows the experimental results of the transductive node classification task in the TI scenario. Some GSL algorithms are designed for TR scenario (i.e., GRCN, IDGL, etc.), so we use kNN graphs as their original graph structure. As we can observe, on the homophily graph datasets, GSL outperforms baselines, such as MLP, $GCN_{knn}$ and $GAT_{knn}$. However, on the heterophily graph datasets, most GSL algorithms often have difficulty achieving better results than baseline models. As mentioned in earlier literature, a network with randomness tends to get better performance utilizing kNN for direct information propagation [17]. Therefore, traditional message-passing neural networks with kNN graphs demonstrate powerful performance. In addition, as observed in the TR scenario, models that leverage self-supervision to extract abundant unlabeled information often achieve better performance.

## 4.3 Performance on heterogeneous graph node-level representation learning

In this section, we evaluate the performance of GSL algorithms on heterogeneous node classification task and use Macro-F1 and Micro-F1 as our evaluation metrics. Table 4 shows the experimental results on heterogeneous graph datasets. By observing the results, we can find that: 1) Because GTN and HGSL consider both heterogeneity and structure learning, they generally outperform other models on heterogeneous graph datasets. 2) GSL algorithms generally outperform the vanilla GNN models (e.g. GCN and GAT) since they have learned better structures to facilitate message passing. 3) Due to the majority of GSL algorithms not explicitly accounting for heterogeneity, they may exhibit poorer performance on heterogeneous graph datasets. 4) Some datasets (e.g. Yelp) exhibit stronger heterogeneity, and on such datasets, models that consider heterogeneity (e.g. HAN, GTN, and HGSL) perform significantly better.

Table 2: Accuracy $\pm$ STD comparison (%) under the standard setting of transductive node classification task in the Topology Refinement (TR) scenario, which means the original graph structure is available for each method. Performance is averaged from 10 independent repetitions. The highest results are highlighted with **bold** , while the second highest results are marked with underline . "OOM" denotes out of memory.

| | Cora 0.81 | Citeseer 0.74 | Pubmed 0.80 | ogbn-arxiv 0.65 | Cornell 0.12 | Texas 0.06 | Wisconsin 0.18 | Actor 0.22 |
|---|---|---|---|---|---|---|---|---|
| Edge Hom. | | | | | | | | |
| GCN | $81.46_{\pm0.58}$ | $71.36_{\pm0.31}$ | $79.18_{\pm0.29}$ | $70.77_{\pm0.19}$ | $47.84_{\pm5.55}$ | $57.83_{\pm2.76}$ | $57.45_{\pm4.30}$ | $30.01_{\pm0.77}$ |
| GAT | $81.41_{\pm0.77}$ | $70.69_{\pm0.58}$ | $77.85_{\pm0.42}$ | $69.90_{\pm0.25}$ | $46.22_{\pm6.33}$ | $54.05_{\pm7.35}$ | $57.65_{\pm7.75}$ | $28.91_{\pm0.83}$ |
| GPRGNN | $83.66_{\pm0.77}$ | $71.64_{\pm0.49}$ | $75.99_{\pm1.63}$ | $50.80_{\pm0.29}$ | **$76.76_{\pm5.30}$** | **$85.14_{\pm3.68}$** | **$83.33_{\pm3.42}$** | **$34.09_{\pm1.09}$** |
| LDS | $83.01_{\pm0.41}$ | **$73.55_{\pm0.54}$** | OOM | OOM | $47.87_{\pm7.14}$ | $58.92_{\pm4.32}$ | $61.70_{\pm3.58}$ | $31.05_{\pm1.31}$ |
| GRCN | $83.87_{\pm0.49}$ | $72.43_{\pm0.61}$ | $78.92_{\pm0.39}$ | OOM | $54.32_{\pm8.24}$ | $62.16_{\pm7.05}$ | $56.08_{\pm7.19}$ | $29.97_{\pm0.71}$ |
| ProGNN | $80.30_{\pm0.57}$ | $68.51_{\pm0.52}$ | OOM | OOM | $54.05_{\pm6.16}$ | $48.37_{\pm12.17}$ | $62.54_{\pm7.56}$ | $22.35_{\pm0.88}$ |
| IDGL | **$83.88_{\pm0.42}$** | $72.20_{\pm1.18}$ | $80.00_{\pm0.38}$ | OOM | $50.00_{\pm8.98}$ | $62.43_{\pm6.09}$ | $59.41_{\pm4.11}$ | $28.16_{\pm1.41}$ |
| GEN | $80.21_{\pm1.72}$ | $71.15_{\pm1.81}$ | $78.91_{\pm0.69}$ | OOM | $57.02_{\pm7.19}$ | $65.94_{\pm1.38}$ | $66.07_{\pm3.72}$ | $27.21_{\pm2.05}$ |
| CoGSL | $81.76_{\pm0.24}$ | $73.09_{\pm0.42}$ | OOM | OOM | $52.16_{\pm3.21}$ | $59.46_{\pm4.36}$ | $58.82_{\pm1.52}$ | $32.95_{\pm1.20}$ |
| SUBLIME | $83.40_{\pm0.42}$ | $72.30_{\pm1.09}$ | **$80.90_{\pm0.94}$** | **$71.75_{\pm0.36}$** | $70.54_{\pm5.98}$ | $77.03_{\pm4.23}$ | $78.82_{\pm6.55}$ | $33.57_{\pm0.68}$ |
| STABLE | $80.20_{\pm0.68}$ | $68.91_{\pm1.01}$ | OOM | OOM | $44.03_{\pm4.05}$ | $55.24_{\pm6.04}$ | $53.00_{\pm5.27}$ | $30.18_{\pm1.00}$ |
| NodeFormer | $80.28_{\pm0.82}$ | $71.31_{\pm0.98}$ | $78.21_{\pm1.43}$ | $55.40_{\pm0.23}$ | $42.70_{\pm5.51}$ | $58.92_{\pm4.32}$ | $48.43_{\pm7.02}$ | $25.51_{\pm1.77}$ |
| GSR | $82.48_{\pm0.43}$ | $71.10_{\pm0.25}$ | $78.09_{\pm0.53}$ | OOM | $44.32_{\pm2.16}$ | $60.81_{\pm4.87}$ | $56.86_{\pm1.24}$ | $30.23_{\pm0.38}$ |

Table 3: Accuracy $\pm$ STD comparison (%) under the standard setting of transductive node classification task in the Topology Inference (TI) scenario, which means the original graph structure is not available for each method.

| | Cora 0.81 | Citeseer 0.74 | Pubmed 0.80 | ogbn-arxiv 0.65 | Cornell 0.12 | Texas 0.06 | Wisconsin 0.18 | Actor 0.22 |
|---|---|---|---|---|---|---|---|---|
| Edge Hom. | | | | | | | | |
| MLP | $58.55_{\pm0.80}$ | $59.52_{\pm0.64}$ | $73.00_{\pm0.30}$ | $55.21_{\pm0.11}$ | $71.35_{\pm6.19}$ | $80.27_{\pm5.93}$ | **$84.71_{\pm3.14}$** | $35.49_{\pm1.04}$ |
| $GCN_{knn}$ | $66.10_{\pm0.44}$ | $68.33_{\pm0.89}$ | $69.23_{\pm0.49}$ | $55.21_{\pm0.22}$ | $75.14_{\pm2.65}$ | $75.95_{\pm4.43}$ | $84.12_{\pm3.97}$ | $32.98_{\pm0.49}$ |
| $GAT_{knn}$ | $64.62_{\pm1.04}$ | $68.05_{\pm1.12}$ | $68.76_{\pm0.80}$ | $55.92_{\pm0.30}$ | $74.05_{\pm5.16}$ | $76.49_{\pm4.99}$ | $82.16_{\pm4.06}$ | $31.67_{\pm1.19}$ |
| $GPRGNN_{knn}$ | $69.27_{\pm0.62}$ | $70.29_{\pm0.54}$ | $68.19_{\pm1.19}$ | $51.39_{\pm0.13}$ | **$75.68_{\pm2.70}$** | **$81.08_{\pm4.18}$** | $84.12_{\pm3.22}$ | $34.71_{\pm1.51}$ |
| LDS | $69.87_{\pm0.41}$ | $72.43_{\pm0.61}$ | OOM | OOM | $72.65_{\pm3.86}$ | $70.20_{\pm5.07}$ | $78.14_{\pm4.50}$ | $32.39_{\pm0.79}$ |
| $GRCN_{knn}$ | $69.48_{\pm0.66}$ | $68.41_{\pm0.50}$ | $68.96_{\pm0.85}$ | OOM | $71.08_{\pm6.84}$ | $74.32_{\pm5.02}$ | $78.63_{\pm4.92}$ | $30.83_{\pm0.76}$ |
| $ProGNN_{knn}$ | $67.11_{\pm0.56}$ | $64.55_{\pm0.95}$ | OOM | OOM | $71.35_{\pm4.04}$ | $71.89_{\pm5.69}$ | $72.94_{\pm7.93}$ | $31.56_{\pm1.14}$ |
| $IDGL_{knn}$ | $69.74_{\pm0.57}$ | $66.33_{\pm0.84}$ | $74.01_{\pm0.64}$ | OOM | $72.70_{\pm4.75}$ | $75.40_{\pm4.75}$ | $79.21_{\pm3.94}$ | $33.07_{\pm1.37}$ |
| $GEN_{knn}$ | $66.95_{\pm1.40}$ | $67.29_{\pm1.17}$ | $69.76_{\pm1.53}$ | OOM | $71.08_{\pm5.54}$ | $74.59_{\pm3.46}$ | $81.76_{\pm2.91}$ | $31.28_{\pm1.06}$ |
| $CoGSL_{knn}$ | $66.65_{\pm0.37}$ | $68.72_{\pm0.84}$ | OOM | OOM | $70.27_{\pm3.42}$ | $72.70_{\pm4.26}$ | $76.96_{\pm5.25}$ | $34.52_{\pm1.56}$ |
| $GSR_{knn}$ | $66.28_{\pm0.59}$ | $66.77_{\pm0.62}$ | $68.49_{\pm1.49}$ | OOM | $70.27_{\pm3.62}$ | $74.86_{\pm3.63}$ | $78.62_{\pm5.91}$ | $33.73_{\pm1.12}$ |
| SLAPS | $72.28_{\pm0.97}$ | $70.71_{\pm1.13}$ | $74.50_{\pm1.47}$ | $55.19_{\pm0.21}$ | $74.59_{\pm3.67}$ | $79.19_{\pm4.99}$ | $81.96_{\pm3.26}$ | **$37.16_{\pm0.91}$** |
| SUBLIME | $72.74_{\pm1.91}$ | **$72.63_{\pm0.60}$** | $75.08_{\pm0.55}$ | $55.57_{\pm0.18}$ | $72.35_{\pm3.57}$ | $75.51_{\pm5.08}$ | $82.14_{\pm2.62}$ | $32.20_{\pm1.02}$ |
| NodeFormer | $54.35_{\pm5.33}$ | $45.90_{\pm5.42}$ | $59.83_{\pm6.50}$ | $55.37_{\pm0.23}$ | $42.70_{\pm5.51}$ | $58.92_{\pm4.32}$ | $48.24_{\pm6.63}$ | $29.24_{\pm1.68}$ |
| HES-GSL | **$73.68_{\pm1.04}$** | $70.12_{\pm1.11}$ | **$77.08_{\pm0.78}$** | **$56.46_{\pm0.27}$** | $66.22_{\pm6.19}$ | $74.05_{\pm6.42}$ | $79.61_{\pm5.28}$ | $36.73_{\pm0.76}$ |

## 4.4 Performance of GSL algorithms on graph-level tasks

In this section, we conduct graph classification experiments on four social datasets (i.e., `IMDB-B`, `RDT-B`, `COLLAB`, and `IMDB-M`) and two biological datasets (i.e., `MUTAG` and `PROTEINS`). Table 5 shows the experimental results of average accuracy and the standard deviation of 10-fold cross-validation. We can observe that HGP-SL (with GCN as the encoder) consistently outperforms GCN on all datasets. However, we find that VIB-GSL exhibits strong instability across different random seeds. And due to the absence of training scripts in the official code[2], we performed hyperparameter tuning based on the parameter search space ($\beta \in \{10^{-1}, 10^{-2}, 10^{-3}, 10^{-4}, 10^{-5}, 10^{-6}\}$) provided in the paper, but we are unable to surpass the performance of the baseline models consistently. Lastly, we conducted an analysis of graph-level GSL algorithms on long-range graph dataset [8]. For detailed information, please refer to Appendix B.

---

[2]https://github.com/RingBDStack/VIB-GSL

Table 4: Macro-F1 and Micro-F1 ± STD comparison (%) under the standard setting of heterogeneous node classification task.

| Method | ACM | | DBLP | | Yelp | |
|---|---|---|---|---|---|---|
| | Macro-F1 | Micro-F1 | Macro-F1 | Micro-F1 | Macro-F1 | Micro-F1 |
| GCN | $90.27_{\pm0.59}$ | $90.18_{\pm0.61}$ | $90.01_{\pm0.32}$ | $90.99_{\pm0.28}$ | $78.01_{\pm1.89}$ | $81.03_{\pm1.81}$ |
| GAT | $91.52_{\pm0.62}$ | $91.46_{\pm0.62}$ | $90.22_{\pm0.37}$ | $91.13_{\pm0.40}$ | $82.12_{\pm1.47}$ | $84.43_{\pm1.56}$ |
| HAN | $91.67_{\pm0.39}$ | $91.47_{\pm0.22}$ | $90.53_{\pm0.24}$ | $91.47_{\pm0.22}$ | $88.49_{\pm1.73}$ | $88.78_{\pm1.40}$ |
| LDS | $92.35_{\pm0.43}$ | $92.05_{\pm0.26}$ | $88.11_{\pm0.86}$ | $88.74_{\pm0.85}$ | $75.98_{\pm2.35}$ | $78.14_{\pm1.98}$ |
| GRCN | $93.04_{\pm0.17}$ | $92.94_{\pm0.18}$ | $88.33_{\pm0.47}$ | $89.43_{\pm0.44}$ | $76.05_{\pm1.05}$ | $80.68_{\pm0.96}$ |
| IDGL | $91.69_{\pm1.24}$ | $91.63_{\pm1.24}$ | $89.65_{\pm0.60}$ | $90.61_{\pm0.56}$ | $76.98_{\pm5.78}$ | $79.15_{\pm5.06}$ |
| ProGNN | $90.57_{\pm1.03}$ | $90.50_{\pm1.29}$ | $83.13_{\pm1.56}$ | $84.83_{\pm1.36}$ | $51.76_{\pm1.46}$ | $58.39_{\pm1.25}$ |
| GEN | $87.91_{\pm2.78}$ | $87.88_{\pm2.61}$ | $89.74_{\pm0.69}$ | $90.65_{\pm0.71}$ | $80.43_{\pm3.78}$ | $82.68_{\pm2.84}$ |
| STABLE | $83.54_{\pm4.20}$ | $83.38_{\pm4.51}$ | $75.18_{\pm1.95}$ | $76.42_{\pm1.95}$ | $71.48_{\pm4.71}$ | $76.62_{\pm2.75}$ |
| GEN | $87.91_{\pm2.78}$ | $87.88_{\pm2.61}$ | $89.74_{\pm0.69}$ | $90.65_{\pm0.71}$ | $80.43_{\pm3.78}$ | $82.68_{\pm2.84}$ |
| SUBLIME | $92.42_{\pm0.16}$ | $92.13_{\pm0.37}$ | $90.98_{\pm0.37}$ | $91.82_{\pm0.27}$ | $79.68_{\pm0.79}$ | $82.99_{\pm0.82}$ |
| NodeFormer | $91.33_{\pm0.77}$ | $90.60_{\pm0.95}$ | $79.54_{\pm0.78}$ | $80.56_{\pm0.62}$ | $91.69_{\pm0.65}$ | $90.59_{\pm1.21}$ |
| GSR | $92.14_{\pm1.08}$ | $92.11_{\pm0.99}$ | $76.59_{\pm0.45}$ | $77.69_{\pm0.42}$ | $83.85_{\pm0.76}$ | $85.73_{\pm0.54}$ |
| GTN | $92.04_{\pm0.38}$ | $91.94_{\pm0.39}$ | $90.52_{\pm0.45}$ | $91.48_{\pm0.39}$ | $92.98_{\pm0.52}$ | $92.44_{\pm0.46}$ |
| HGSL | $93.23_{\pm0.50}$ | $93.13_{\pm0.51}$ | $91.58_{\pm0.40}$ | $92.49_{\pm0.35}$ | $92.79_{\pm0.44}$ | $92.24_{\pm0.48}$ |

Table 5: Accuracy ± STD comparison (%) under the setting of graph-level classification task.

| Method | IMDB-B | RDT-B | COLLAB | IMDB-M | MUTAG | PROTEINS |
|---|---|---|---|---|---|---|
| GCN | $73.20_{\pm4.29}$ | $70.10_{\pm5.80}$ | $76.96_{\pm2.28}$ | $49.85_{\pm3.84}$ | $73.92_{\pm8.84}$ | $67.52_{\pm6.71}$ |
| VIB-GSL (GCN) | $71.90_{\pm4.48}$ | $68.95_{\pm2.66}$ | $77.14_{\pm1.59}$ | $49.05_{\pm5.52}$ | $68.63_{\pm5.15}$ | $65.68_{\pm8.53}$ |
| HGP-SL (GCN) | $74.10_{\pm4.55}$ | OOM | $78.06_{\pm2.17}$ | $51.07_{\pm2.00}$ | $78.07_{\pm10.85}$ | $70.80_{\pm4.25}$ |
| GAT | $72.30_{\pm2.26}$ | $73.55_{\pm4.76}$ | $79.08_{\pm1.36}$ | $48.90_{\pm2.98}$ | $78.71_{\pm7.51}$ | $68.63_{\pm6.24}$ |
| VIB-GSL (GAT) | $72.10_{\pm5.69}$ | OOM | $77.54_{\pm1.85}$ | $49.06_{\pm4.55}$ | $77.13_{\pm9.95}$ | $67.09_{\pm8.43}$ |
| SAGE | $72.60_{\pm3.69}$ | $70.20_{\pm4.11}$ | $75.58_{\pm2.04}$ | $48.55_{\pm2.03}$ | $68.65_{\pm4.31}$ | $64.47_{\pm7.15}$ |
| VIB-GSL (SAGE) | $73.00_{\pm4.78}$ | $65.75_{\pm3.17}$ | $77.74_{\pm1.52}$ | $48.79_{\pm5.06}$ | $72.81_{\pm11.41}$ | $66.61_{\pm4.48}$ |
| HGP-SL (SAGE) | $71.50_{\pm5.24}$ | OOM | $78.64_{\pm1.47}$ | $49.67_{\pm3.09}$ | $77.13_{\pm3.29}$ | $73.32_{\pm2.06}$ |
| GIN | $73.00_{\pm2.67}$ | $71.70_{\pm5.01}$ | $79.86_{\pm1.64}$ | $50.30_{\pm3.52}$ | $87.19_{\pm8.05}$ | $69.07_{\pm5.62}$ |
| VIB-GSL (GIN) | $69.90_{\pm3.90}$ | $75.85_{\pm3.63}$ | $77.25_{\pm2.34}$ | $49.97_{\pm3.65}$ | $85.18_{\pm10.11}$ | $75.15_{\pm5.72}$ |
| HGP-SL (GIN) | $73.50_{\pm6.25}$ | OOM | $80.14_{\pm1.51}$ | $48.67_{\pm2.58}$ | $73.92_{\pm6.24}$ | $69.37_{\pm3.95}$ |

## 4.5 Robustness analysis of GSL algorithms

To investigate the robustness of GSL algorithms, we primarily focus on three aspects: structure robustness, feature robustness, and supervision signal robustness. Due to limited space, we predominantly investigate the transductive node classification task in our paper. Nevertheless, researchers can utilize our GSLB library to efficiently and conveniently conduct experiments on other tasks as well.

**Robustness analysis with respect to different supervision signals.** We have discovered that GSL maintains surprising performance in scenarios with a scarcity of labeled samples. We varied the number of labels per class in the Cora and Citeseer datasets and selected two baseline models, DAGNN [25] and Self-Training [22], that performed well in scenarios with limited labels. As shown in Fig-

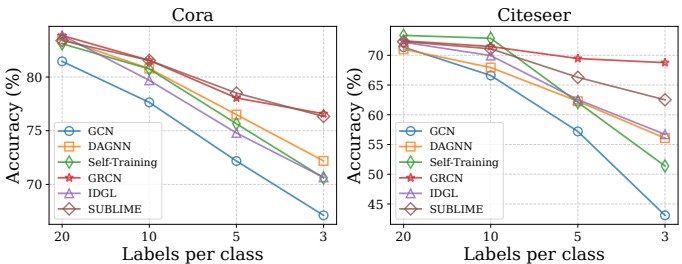

Figure 2: Performance of baselines and GSL algorithms with respect to different numbers of labels per class on Cora and Citeseer.

Table 6: Accuracy $\pm$ STD comparison (%) with respect to different perturbation rates. Jaccard and SimPGCN are representative state-of-the-art defense GNNs.

| Dataset | Ptb Rate | GCN | Jaccard | SimPGCN | IDGL | GRCN | ProGNN | STABLE | SUBLIME |
|---|---|---|---|---|---|---|---|---|---|
| Cora | 0% | 83.68±0.37 | 83.78±0.50 | 82.66±0.48 | **84.69±1.13** | 84.43±0.26 | 84.53±0.89 | 83.70±0.30 | 83.84±0.28 |
| | 5% | 80.61±0.39 | 81.44±0.48 | 80.35±0.82 | **82.56±0.24** | 81.34±0.50 | 81.47±0.44 | 81.52±0.85 | 79.93±0.58 |
| | 10% | 74.38±0.59 | 75.90±0.64 | 76.50±1.12 | 78.06±0.62 | 77.12±0.38 | 72.61±0.73 | 78.64±1.82 | **78.71±0.46** |
| | 15% | 65.17±0.99 | 77.14±0.70 | 73.77±1.88 | 76.88±0.44 | 73.74±0.61 | 65.68±1.97 | **79.70±1.71** | 79.34±0.61 |
| | 20% | 61.98±1.23 | 70.71±0.91 | 69.08±2.78 | 67.19±0.69 | 69.54±0.58 | 61.07±0.61 | **76.44±2.47** | 75.25±1.08 |
| Citeseer | 0% | **76.56±0.36** | 74.34±0.26 | 74.35±0.74 | 73.87±0.70 | 76.34±0.11 | 73.36±1.52 | 72.65±1.36 | 73.34±1.17 |
| | 5% | 72.51±0.30 | 70.01±0.79 | 72.99±1.05 | 72.46±0.47 | **74.66±0.27** | 71.46±0.47 | 69.66±0.95 | 72.63±0.50 |
| | 10% | 71.92±0.68 | 70.28±1.30 | 72.68±0.54 | 69.72±0.59 | **74.06±0.43** | 69.03±0.60 | 72.79±0.71 | 73.02±0.29 |
| | 15% | 64.44±0.53 | 67.13±1.28 | 71.74±1.46 | 62.83±1.28 | 66.46±1.12 | 65.42±1.20 | 70.98±0.61 | **73.90±0.52** |
| | 20% | 57.51±1.03 | 67.82±0.74 | 70.06±1.86 | 61.16±0.99 | 69.42±1.14 | 57.51±0.36 | 71.90±1.12 | **72.55±0.62** |
| Polblogs | 0% | 95.62±0.69 | 94.93±0.28 | 94.50±0.43 | 94.83±0.20 | **95.65±0.28** | 94.84±0.19 | 95.63±0.32 | 95.27±0.51 |
| | 5% | 80.57±0.66 | 78.17±0.55 | 76.02±1.14 | 79.62±0.65 | **93.70±0.18** | 92.36±0.42 | 89.41±1.63 | 93.24±1.50 |
| | 10% | 71.83±2.37 | 71.86±1.34 | 70.12±1.10 | 74.54±0.69 | 87.99±1.56 | 84.66±0.52 | 89.87±0.82 | **93.62±0.50** |
| | 15% | 66.38±2.17 | 69.93±0.66 | 64.19±1.55 | 75.53±0.83 | 71.85±1.58 | 77.38±0.51 | 89.94±0.89 | **94.29±0.27** |
| | 20% | 68.19±2.24 | 69.22±0.34 | 63.64±1.41 | 71.63±0.62 | 71.73±1.58 | 73.57±0.29 | 87.42±0.69 | **92.60±0.72** |

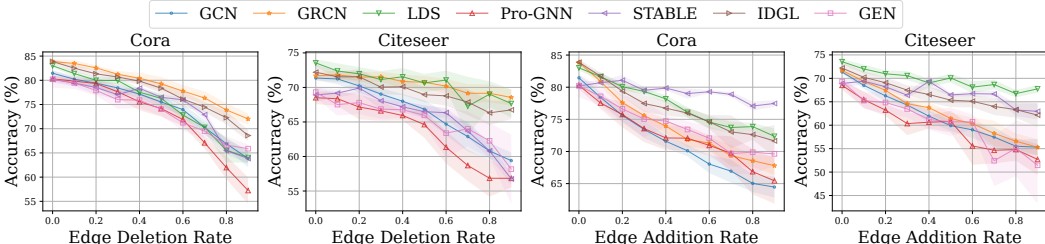

Figure 3: Analysis of robustness when injecting random noise on Cora and Citeseer.

ure 2, we can observe that GSL algorithms (for the sake of brevity, we opted to select only three models: GRCN, IDGL, and SUBLIME) achieve the best results in scenarios with fewer labels available. We speculate that this may be because the learned graph structure is denser and exhibits cleaner community boundaries. As a result, the supervision signals can propagate more effectively within such a structure.

**Robustness analysis with respect to random noise.** We randomly remove edges from or add edges to the original graph structures of Cora and Citeseer, then evaluated the performance of GSL algorithms on the corrupted graphs. We change the ratios of modified edges from 0 to 0.9 to simulate different attack intensities. As shown in Figure 3, as the noise intensity increases, the models' performance generally exhibits a downward trend. And we can observe that GSL algorithms commonly demonstrate a certain degree of robustness, as they tend to exhibit more stable performance than GCN when random noise is injected. Besides, we also found that, due to variations in the graph modeling process, different algorithms display varying levels of robustness when facing edge deletion and edge addition scenarios. For example, GRCN demonstrates strong robustness in edge deletion scenarios. However, in the edge addition scenarios, it only exhibits slight performance improvements compared to GCN. On the contrary, STABLE exhibits strong robustness in the edge deletion scenario, while showing the opposite trend in edge addition.

**Robust analysis with respect to graph topology attack.** Following [21, 55], we conduct robust analysis on three graph datasets, i.e., Cora, Citeseer, and Polblogs. First, we select the largest connected component in the graph, and utilize Mettack [58], a non-targeted adversarial topology attack method, to generate perturbed graphs. We select the perturbation rate from 0% to 20%. Table 6 shows the performance of GSL algorithms on three datasets with respect to various perturbation rates. Surprisingly, we can observe that most GSL algorithms exhibit strong robustness against graph topology attacks, even better than state-of-the-art defense GNNs (e.g., Jaccard [41] and SimPGCN [19]). GSL can effectively remove the newly added adversarial edges, and recover important edges to promote message passing. As mentioned in Li et al. [21], optimizing graph structures based on either features or supervised signals might not be reliable. We found that self-supervised graph structure modeling methods (e.g., STABLE and SUBLIME) show excellent

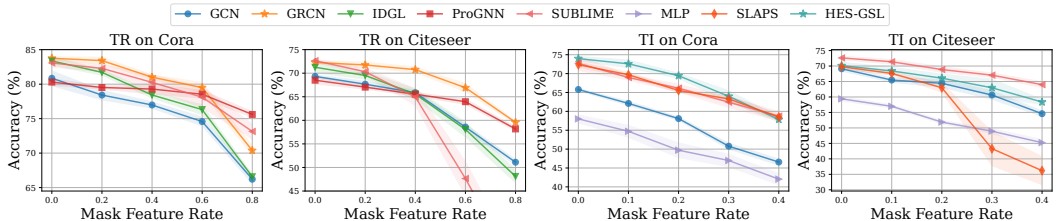

Figure 4: Analysis of robustness when injecting random feature noise on Cora and Citeseer.

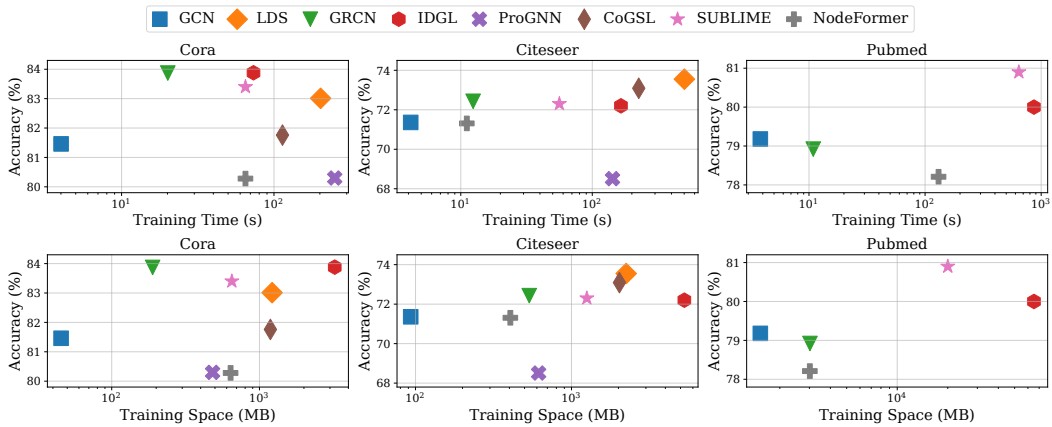

Figure 5: Training time and space analysis on `Cora`, `Citeseer` and `Pubmed`.

performance on corrupted graph structure datasets, which means unsupervised representation learning might produce more reliable and high-quality representations to conduct structure modeling.

**Robust analysis with respect to feature noise.** On the basis of exploring structural robustness, we also study the feature robustness of GSL. We randomly mask a certain proportion of node features by filling them with zeros, to investigate the performance of GSL algorithms when node features are subjected to varying degrees of damage. As shown in Figure 4, we can observe that: 1) the node features play a more critical role than the structure on certain datasets. Under the same noise degree, feature noise brings more performance degradation compared with structure noise; 2) Interestingly, while most existing GSL methods rely on feature similarity between pairs of nodes to learn graph structure, they still exhibit good robustness when facing noisy node features; 3) edge-oriented algorithms (e.g., ProGNN) show stronger feature robustness, because they optimize adjacency matrix directly, and have less dependence on pairs of node features.

### 4.6 Efficiency and scalability analysis

In this section, we analyze the efficiency and scalability of GSL algorithms on Cora, Citeseer, and Pubmed datasets. For time efficiency, we evaluate the efficiency of the algorithms by measuring the time it takes for them to converge, i.e., achieve the best performance on the validation set. For scalability, we set all models to their dense version to ensure a fair comparison. As shown in Figure 5, GSL algorithms generally have higher time and space complexity compared to GCN. This limitation restricts the application of GSL on large-scale graphs. We can observe that some algorithms (e.g., GRCN) can achieve relatively good performance improvement with less complexity increase. Besides, although some algorithms (e.g., IDGL, LDS, and SUBLIME) achieve remarkable effectiveness improvement, they largely increase the complexity of time and space.

## 5 Conclusion and Future Directions

In this paper, we give a brief introduction and overview of graph structure learning. Then we present the first Graph Structure Learning Benchmark (GSLB) consisting of 16 algorithms and 20 datasets

for various tasks. Based on GSLB, we conducted extensive experiments to reveal and analyze the performance of GSL algorithms in different scenarios and tasks. Through our comparative study, we find that GSL achieves promising results in heterophily, robustness, etc. The goal of this work is to understand the current state of development of GSL and provide insights for future research.

Notwithstanding the promising results that have been made, there are still some critical challenges and research directions worthy of future investigation.

- **Insufficient scalability**. Most existing works model the existence probability of edges based on node pairs, with a complexity of $O(N^2)$. This makes it challenging to employ GSL in large-scale graphs in real-world applications. Future work should focus on overcoming the limitations of GSL in terms of complexity.

- **Surprising performance with few labels**. We have observed that GSL learns denser and more distinct graph structures, which facilitates the propagation of supervision signals. Most existing GNNs that address few label problem are based on deep GNNs [25, 13] or semi-supervised approaches [6, 35, 20], without refining the graph structure. In the future, it would be worth exploring the combination of increasing the supervision signals and making the graph structure more suitable for propagating those signals.

- **Excellent performance of unsupervised GSL in robustness**. Some algorithms using self-supervised methods for learning graph structures exhibit excellent performance in robustness, which may be attributed to the avoidance of unreliable supervision signals. In the future, further exploration can be done to utilize unsupervised structure learning for designing defense models.

- **Hard to apply on incomplete graphs.** Most existing algorithms rely on pairwise node embeddings to generate the probability of edge existence. The underlying assumption is that all attributes of nodes on the graph are complete. However, it is common in practice that some nodes or all nodes have no features. Future research should address the challenges of structure learning on incomplete graphs.

## Acknowledge

This work was partially supported by the Research Grants Council of Hong Kong, No. 14202919 and No. 14205520.

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
