# Appendix A  Datasets and Algorithms

## A.1  Datasets

All of the public datasets used in our benchmark were previously published, either as graph representation learning benchmarks or new datasets for specific graph tasks. The datasets cover various downstream tasks and a multitude of domains: citation network, social network, bioinformatics, website networks, computer vision, and co-occurrence network. Table 7, Table 8, and Table 9 provides the detailed statistics about diverse datasets. We adopt the following benchmark datasets since i) they are widely applied to develop and evaluate GNN models; ii) they contain diverse graph properties from small-scale to large-scale, from homogeneous to heterogeneous, from homophilic to heterophilic, or from node-level to graph-level. The detailed descriptions of these datasets are listed in the following:

- **Cora, Citeseer, Pubmed.** They are the scientific citation network datasets [47], where nodes and edges represent the scientific publications and their citation relationships, respectively. Each publication in the dataset is described by a 0/1-valued word vector indicating the absence/presence of the corresponding word from the dictionary. Each node is associated with a one-hot label, where the node classification task is to predict which class the corresponding publication belongs to.

- **Ogbn-arxiv.** The ogbn-arxiv dataset is a benchmark citation network collected in open graph benchmark (OGB) [15], which consists of a large number of nodes and edges, and has been widely utilized to evaluate GNN models[3]. Each node represents an arXiv paper from the computer science domain, and each directed edge indicates that one paper cites another one. The node is described by a 128-dimensional word embedding extracted from the title and abstract in the corresponding publication.

- **Cornell, Texas, Wisconsin.** They are the website datasets from WebKB[4], collected from computer science departments of various universities by Carnegie Mellon University [34]. While nodes represent webpages in the webpage datasets, edges are hyperlinks between them. The node feature vectors are given by bag-of-word representation of the corresponding webpages. Each node is associated with a one-hot label to indicate one of the following five categories, i.e., student, project, course, staff, and faculty.

- **Actor.** This is an actor co-occurrence network, which is an actor-only induced subgraph of the film-director-actor-writer network [37]. In the co-occurrence network, nodes correspond to actors and edges denote the co-occurrence relationships on the same Wikipedia pages. Node feature vectors are described by the bag-of-word representation of keywords in the actors' Wikipedia pages.

- **Polblogs.** This is a blog network, which consists of 1,222 vertexes and 16,716 edges. Each node represents a blog page, and each edge denotes a hyperlink between pages. Every blog has a political attribute: conservative or liberal, which is the label of each node.

- **ACM, DBLP, Yelp.** They are real-world heterogeneous graph datasets. DBLP and ACM [49] are citation networks, where DBLP contains three types of nodes (papers (P), authors (A), conferences (C)), four types of edges (PA, AP, PC, CP), and research areas of authors as labels; ACM contains three types of nodes (papers (P), authors (A), subjects (S)), four types of edges (PA, AP, PS, SP), and categories of papers as labels. Yelp [29] is a review dataset and contains three types of nodes (businesses (B), users (U), services (S)), and 9rating levels (L). The business nodes are labeled by their category.

- **IMDB-B, IMDB-M, REDDIT-B, COLLAB.** IMDB-BINARY and IMDB-MULTI are movie collaboration datasets that consist of the ego-networks of 1,000 actors/actresses who played roles in movies in IMDB. In each graph, nodes represent actors or actresses, and there is an edge between them if they appear in the same movie. REDDIT-BINARY consists of graphs corresponding to online discussions on Reddit. In each graph, nodes represent users, and there is an edge between them if at least one of them responds to the other's comment. COLLAB is a scientific collaboration dataset. Each graph corresponds to a researcher's ego network, i.e., the researcher and its collaborators are nodes and an edge denotes collaboration between two researchers.

---

[3]https://ogb.stanford.edu
[4]http://www.cs.cmu.edu/afs/cs. cmu.edu/project/theo-11/www/wwkb

- **PROTEINS, MUTAG.** PROTEINS represents macromolecules and it was derived from Dobson and Doig [7]. The task of it is to predict whether a protein is an enzyme. Nodes represent the amino acids and there is an edge if they are less than 6 Angstroms apart. MUTAG is a collection of nitroaromatic compounds where nodes stand for atoms and edges between nodes represent bonds between the corresponding atoms.

- **Peptides-func, Peptides-struct.** They are proposed by recent long-range graph benchmark [8] for exploring the ability of GNNs to capture long-range dependencies. Each graph in datasets is a peptide (a short chain of amino acids), while nodes correspond to the heavy (non-hydrogen) atoms and edges represent the bonds between them. Peptides-func is a multi-label graph classification dataset. Graphs are divided into 10 classes based on the peptide functions. Peptides-struct is a multi-label graph regression dataset based on the 3D structure of the peptides.

Table 7: Statistics of homogeneous node classification datasets.

| Datasets | Cora | Citeseer | Pubmed | ogbn-arxiv | Cornell | Texas | Wisconsin | Actor | Polblogs |
|---|---|---|---|---|---|---|---|---|---|
| #Nodes | 2,708 | 3,327 | 19,717 | 169,343 | 183 | 183 | 251 | 7,600 | 1,222 |
| #Edges | 5,278 | 4,614 | 44,325 | 1,157,799 | 277 | 279 | 450 | 26,659 | 16,716 |
| #Classes | 7 | 6 | 3 | 40 | 5 | 5 | 5 | 5 | 2 |
| #Features | 1,433 | 3,703 | 500 | 767 | 1,703 | 1,703 | 1,703 | 932 | — |
| #Homophily | 0.81 | 0.74 | 0.80 | 0.65 | 0.12 | 0.06 | 0.18 | 0.22 | 0.91 |
| Avg. #Degree | 3.90 | 2.77 | 4.50 | 13.67 | 3.03 | 3.05 | 3.59 | 7.02 | 27.36 |

Table 8: Statistics of heterogeneous node classification datasets.

| Datasets | #Nodes | #Edge | #Node Type | #Edge Type | #Features | Avg. #Degree | #Training | #Validation | #Test |
|---|---|---|---|---|---|---|---|---|---|
| ACM | 8,994 | 25,922 | 3 | 4 | 1,902 | 1.51 | 600 | 300 | 2,125 |
| DBLP | 7,305 | 19,816 | 3 | 4 | 334 | 1.44 | 600 | 300 | 2,057 |
| Yelp | 3,913 | 77,176 | 3 | 6 | 82 | 3.72 | 300 | 300 | 2,014 |

Table 9: Statistics of graph-level datasets.

| Datasets | IMDB-B | IMDB-M | REDDIT-B | COLLAB | PROTEINS | MUTAG | Peptides-func | Peptides-struct |
|---|---|---|---|---|---|---|---|---|
| #Graphs | 1,000 | 1,500 | 2,000 | 5,000 | 1,113 | 188 | 15,535 | 15,535 |
| #Classes | 2 | 3 | 2 | 3 | 2 | 2 | 10 | — |
| Avg. #Nodes | 19.8 | 13.0 | 429.7 | 74.5 | 39.1 | 17.9 | 150.9 | 150.9 |
| Avg. #Edges | 96.5 | 65.9 | 497.8 | 2,457.5 | 72.8 | 19.8 | 307.3 | 307.3 |
| Avg. #Degree | 4.9 | 5.1 | 1.2 | 33.0 | 1.9 | 1.1 | 2.0 | 2.0 |
| Avg. #Diameter | 1.9 | 1.5 | — | 1.9 | — | 8.2 | 27.6 | 27.6 |
| Vertex labels | ✘ | ✘ | ✘ | ✘ | ✔ | ✔ | ✘ | ✘ |
| Task | Class. | Class. | Class. | Class. | Class. | Class. | Class. | Regre. |
| Domain | Social | Social | Social | Social | Biochemical | Biochemical | Biochemical | Biochemical |

## A.2 Algorithms

In our developed GSLB, we integrate 16 state-of-the-art GSL algorithms, including 12 homogeneous node classification models: LDS, GRCN, ProGNN, IDGL, GEN, CoGSL, SLAPS, SUBLIME, STABLE, NodeFormer, GSR, HES-GSL; 2 heterogeneous node classification models: GTN and HGSL; 2 graph-level models: VIB-GSL and HGP-SL. Each is a representative algorithm in its respective task and covers both the early and recent work of GSL. In order to better organize and understand the GSL, we demonstrate a high-level comparison of existing representative algorithms in Table 10.

- **LDS** [11]. LDS is an early work in GSL. It proposes to approximately solving a bilevel program to jointly learn the graph structure and parameters of GNNs.

- **GRCN** [48]. GRCN designs a graph revision module to predict missing edges and revise edge weights. To reduce the complexity of GRCN, Fast-GRCN only calculates the similarity matrix in the first epoch and then computes the values of the kNN-sparse matrix for the remaining epochs. Therefore, the complexity of GRCN can be reduced to $O(NK)$, where $K$ is the number of top-$K$ important neighbors of each node.

- **ProGNN** [18]. ProGNN treats the adjacency matrix as a learnable variable and directly optimizes it with GNNs to learn a robust structure.

Table 10: Summary of representative Graph Structure Learning (GSL) methods. *Task* refers to the downstream task that the corresponding method is applicable for. For *Oriented*, 'Node' means deriving the edge connectivity based on pairwise node embeddings, and 'Edge' means directly optimizing the graph adjacency matrix. *Requirement* means the required input data to models for training. For *Graph Regularization*, 'SP' means sparsity, 'SM' means smoothness, 'CON' means connectivity, and 'CP' means community preservation.

| | Method | Task | Oriented | Requirement | | Structure Modeling | Graph Regularization | | | | Complexity | Code |
|---|---|---|---|---|---|---|---|---|---|---|---|---|
| | | | | Structure | Labels | | SP | SM | CON | CP | | |
| Metric | GRCN [48] | NC | Node | ✓ | ✓ | Inner product | ✓ | | | | $O(N^2)$ or $O(NK)$ | Link |
| | IDGL [4] | NC | Node | ✓ | ✓ | Cosine similarity | ✓ | ✓ | ✓ | | $O(N^2)$ or $O(Nm)$ | Link |
| | HGSL [53] | HNC | Node | ✓ | ✓ | Cosine similarity | ✓ | | | | $O(N^2)$ | Link |
| | HGP-SL [52] | GC | Node | ✓ | ✓ | Attention | ✓ | | | | $O(N^2)$ | Link |
| | GEN [39] | NC | Node | ✓ | ✓ | Cosine similarity | | | | | $O(N^2)$ | Link |
| | STABLE [21] | NC | Node | ✓ | | Cosine similarity | | | | | $O(N^2)$ | Link |
| | NodeFormer [43] | NC | Node | | ✓ | Attention | ✓ | | | | $O(N)$ or $O(E)$ | Link |
| | GSR [54] | NC | Node | ✓ | | Cosine similarity | | | | | $O(NB)$ | Link |
| | HES-GSL [42] | NC | Node | | ✓ | Cosine similarity | | | | | $O(N^2)$ | Link |
| Neural | GLCN [16] | NC | Node | | ✓ | One-layer neural net | ✓ | ✓ | | | $O(N^2)$ | Link |
| | PTDNet [30] | NC | Node | ✓ | ✓ | Multilayer perceptron | ✓ | | | ✓ | $O(E)$ | Link |
| | VIB-GSL [36] | GC | Node | ✓ | ✓ | Multilayer perceptron | ✓ | | | | $O(N^2)$ | Link |
| | NeuralSparse [55] | NC | Node | ✓ | ✓ | Multilayer perceptron | | | | | $O(E)$ | — |
| | GTN [49] | HNC | Edge | ✓ | ✓ | Convolutional layers | | | | | $O(N^2)$ | Link |
| | CoGSL [26] | NC | Node | ✓ | ✓ | Multilayer perceptron | | | | | $O(N^2)$ | Link |
| Direct | GLNN [12] | NC | Edge | ✓ | ✓ | Free variables | ✓ | ✓ | | | $O(N^2)$ | — |
| | LDS [11] | NC | Edge | | ✓ | Free variables | ✓ | | | | $O(N^2)$ | Link |
| | LRGNN [44] | NC | Edge | ✓ | ✓ | Free variables | | ✓ | | ✓ | $O(N^2)$ | — |
| | ProGNN [18] | NC | Edge | ✓ | ✓ | Free variables | ✓ | ✓ | | ✓ | $O(N^2)$ | Link |
| Hybird | SLAPS [10] | NC | — | | ✓ | Multiple learners | | | | | $O(N^2)$ | Link |
| | SUBLIME [28] | NC | — | | | Multiple learners | | | | | $O(N^2)$ | Link |

- **IDGL** [4]. IDGL jointly and iteratively learns graph structures and graph embeddings. In addition, it also provides a scalable version, namely IDGL-ANCH, which randomly samples $m$ anchors from the node set for each node to calculate affinity scores.

- **GEN** [39]. GEN designs a structure model characterizing the underlying graph generation and an observation model injecting multi-order neighborhood information to accurately infer the graph structure based on Bayesian inference.

- **CoGSL** [26]. CoGSL utilizes mutual information compression to extract compact and robust graph structure, namely "minimal sufficient structure", which maximizes the information about downstream tasks.

- **SLAPS** [10]. SLAPS focuses on topology inference task and provides more supervision signals for inferring a graph structure through self-supervision learning.

- **SUBLIME** [28]. To prevent the reliance on labels, bias of edge distribution, and the limitation on application tasks, SUBLIME learns graph structure in an unsupervised manner. It first generates a learning target graph, anchor graph, and maximizes the agreement between the anchor graph and the learned graph by contrastive learning. It also designs a variety of graph learners and post-processors.

- **STABLE** [21]. STABLE optimizes the learned graph structure through an unsupervised pipeline to avoid using unreliable supervision signals.

- **NodeFormer** [43]. NodeFormer introduces a novel all-pair message-passing scheme for efficiently propagating node signals between arbitrary nodes. Because of the high complexity of Transformer-style architecture, it uses a kernelized Gumbel-Softmax operator to reduce the complexity to linearity.

- **GSR** [54]. GSR first estimates the underlying graph structure by a multi-view contrastive learning framework and then fine-tunes a GNN on the learned structure.

- **HES-GSL** [42]. HES-GSL proves that the task-specific supervision signals may be insufficient to support the learning of both graph structure and parameters of GNNs. Therefore, it proposes homophily-enhanced self-supervision for GSL to provide more supervision information for topology inference.

Table 11: Average Precision (AP) ± STD comparison (%) for `Peptides-func` and Mean Absolute Error (MAE) ± STD comparison for `Peptides-struct`. Each result was obtained from 3 repeated experiments with different random seeds. ↑ represents the larger, the better while ↓ represents the smaller, the better.

| Method | Peptides-Func (AP ↑) | Peptides-Struct (MAE ↓) |
|---|---|---|
| GCN | $58.31_{\pm 0.28}$ | $0.3566_{\pm 0.0014}$ |
| GAT | $45.59_{\pm 0.35}$ | $0.4375_{\pm 0.0036}$ |
| GIN | $\mathbf{60.92}_{\pm \mathbf{0.88}}$ | $0.3647_{\pm 0.0028}$ |
| SAGE | $59.41_{\pm 0.57}$ | $0.3716_{\pm 0.0011}$ |
| VIB-GSL (GCN) | $52.05_{\pm 2.40}$ | $0.3089_{\pm 0.0036}$ |
| VIB-GSL (GAT) | $42.64_{\pm 0.33}$ | $0.3992_{\pm 0.0129}$ |
| VIB-GSL (GIN) | $47.61_{\pm 0.93}$ | $0.3032_{\pm 0.0011}$ |
| VIB-GSL (SAGE) | $54.11_{\pm 0.74}$ | $0.3063_{\pm 0.0014}$ |
| HGP-SL (GCN) | $52.16_{\pm 0.94}$ | $0.2935_{\pm 0.0104}$ |
| HGP-SL (GIN) | $55.30_{\pm 0.88}$ | $\mathbf{0.2786}_{\pm \mathbf{0.0018}}$ |
| HGP-SL (SAGE) | $53.48_{\pm 0.71}$ | $\underline{0.2794}_{\pm 0.0058}$ |

- **GTN** [49]. GTN learns a soft selection of edge types and composite relations for generating useful multi-hop connections for heterogeneous graphs.

- **HGSL** [53]. HGSL firstly attempts to learn heterogeneous graph structure and GNNs jointly. It considers the feature similarity by generating a feature similarity graph and optimizes the complex heterogeneous interactions by generating a feature propagation graph and semantic graph.

- **HGP-SL** [52]. HGP-SL is a graph-level model. It adaptively selects a subset of nodes to form an induced subgraph and utilizes structure learning to refine subgraphs at each layer.

- **VIB-GSL** [36]. VIB-GSL firstly attempts to advance the Information Bottleneck (IB) principle for graph structure learning and proposes to use dot-product self-attention to refine dynamic connections.

## Appendix B    Additional Experimental Results

### B.1    Performance on long-range datasets

Whether GSL can capture long-range dependencies is an interesting topic. Traditional message-passing GNNs simply rely on local neighbors to produce node representations and are hard to learn higher-order information. Recently, LRGB [8] presents a series of graph learning datasets, which arguably require the ability of long-range interactions to achieve strong performance. Table 11 shows the experimental results of graph-level GSL algorithms on two long-range datasets. We can observe that VIB-GSL and HGP-SL do not exhibit promising results on `Peptides-Func` dataset, but show evidently better results on `Peptides-Struct` dataset than the baseline model. We suspect that this may be related to whether the GSL algorithm is suitable for specific downstream tasks. We only investigate the long-range capability of graph-level GSL models. The investigation of long-range capability at the node-level is our future work.

### B.2    Visualiuzation

In order to more intuitively understand the ability and characteristics of GSL, we visualize the original graph structure of Cora and the learned graphs of various GSL algorithms in Figure 6. We select four categories of Cora and randomly sample 10 labeled nodes and 10 unlabeled nodes to extract a subgraph. The elements inside the red rectangle are the intra-class connections, and the elements on the diagonal are self-loops. We can observe that i) most of the learned structures are much denser than the original structure (especially the learned structure of IDGL); ii) in the TI scenarios, SLAPS and HES-GSL will prefer to connect labeled nodes. Moreover, we also make statistics on the properties of the original structure and the learned structures. As shown in Table 12, we list node homophily ratio [34], edge homophily ratio [56], class insensitive edge homophily ratio [24] of the structures. They can be calculated as follows:

Table 12: Graph property statistics of the original and the learned graphs on Cora. *Node Homo.* represents node homophily ratio; *Edge Homo.* represents edge homophily ratio; *EI Homo.* represents class insensitive edge homophily ratio; *Assor.* represents the degree assortativity coefficient.

| Method | Node Homo. | Edge Homo. | EI Homo. | Assor. | Density |
|---|---|---|---|---|---|
| Original | 0.8252 | 0.8100 | 0.7657 | -0.0659 | 0.0014 |
| LDS | 0.7636 | 0.7500 | 0.6842 | -0.0571 | 0.0019 |
| GRCN | 0.5948 | 0.5941 | 0.4857 | -0.0720 | 0.0816 |
| ProGNN | 0.2208 | 0.2041 | 0.0309 | -0.0246 | 0.5666 |
| IDGL | 0.1796 | 0.1796 | 0.0000 | / | 1.0000 |
| GEN | 0.7425 | 0.7612 | 0.7055 | 0.0560 | 0.0085 |
| CoGSL | 0.7300 | 0.7081 | 0.6343 | 0.0469 | 0.0180 |
| SUBLIME | 0.1796 | 0.1796 | 0.0000 | / | 1.0000 |
| STABLE | 0.5259 | 0.5228 | 0.4049 | -0.0153 | 0.0057 |
| NodeFormer | 0.1796 | 0.1796 | 0.0000 | / | 1.0000 |
| SLAPS | 0.5683 | 0.5755 | 0.4820 | 0.0709 | 0.0107 |
| HESGSL | 0.6353 | 0.6635 | 0.5835 | 0.0989 | 0.0171 |

$$\text{Node:} \quad h_{node} = \frac{1}{N} \sum_{v \in \mathcal{V}} \frac{\{(u,v) : u \in \mathcal{N}(v) \wedge y_v = y_u\}}{|\mathcal{N}(v)|}$$

$$\text{Edge:} \quad h_{edge} = \frac{|\{(v,u) : (v,u) \in \mathcal{E} \wedge y_v = y_u\}|}{|\mathcal{E}|}$$

$$\text{EI:} \quad h_{ei} = \frac{1}{C-1} \sum_{k=1}^{C} \max\left(0, h_{edge}^k - \frac{|\mathcal{C}_k|}{|\mathcal{V}|}\right)$$

where $\mathcal{V}$ and $\mathcal{E}$ denotes the set of nodes and edges, $C$ is the number of classes, $|\mathcal{C}_k|$ denotes the number of nodes of class $k$, and $h_{edge}^k$ denotes the edge homophily ratio of class $k$. And we also present the degree assortativity coefficient [32] of structures, which refers to the tendency of nodes to connect with other similar nodes over dissimilar nodes. According to Table 12, we can observe that the learned graph structures do not increase the homophily ratio, thus the homophily ratio may not be the main reason for the improved performance of GSL. We also find that all GSL algorithms make the graph structure mode dense, which indicates that real-world datasets may be too sparse. Finally, we plot the degree distribution of the original graph of Cora and the learned graphs by various GSL algorithms in Figure 7. We can observe that the learned graph structure still follows the long-tail distribution. How to use the GSL to alleviate the unfairness of the classification performance of nodes with different degrees is also a problem worth exploring.

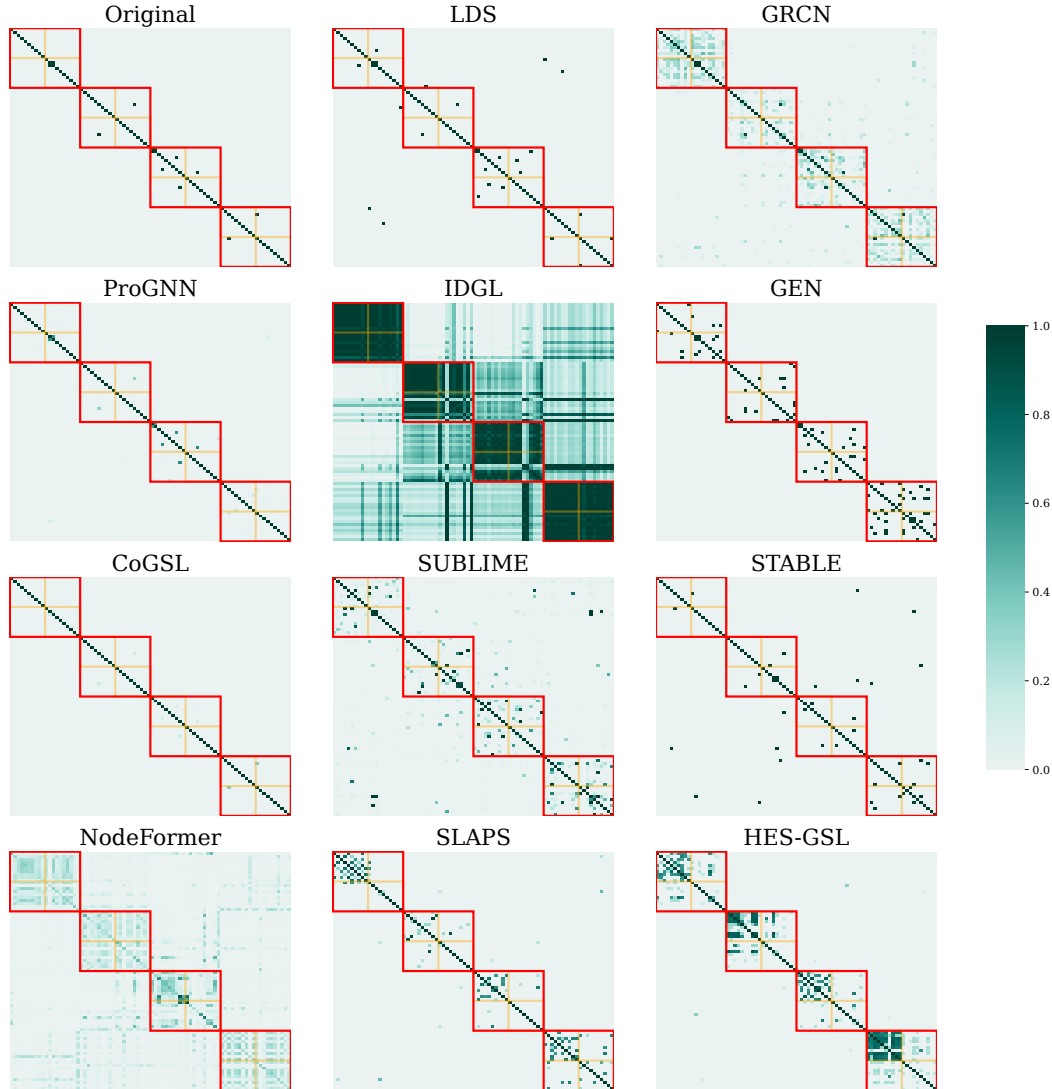

Figure 6: Visualization of the original graph of Cora and the learned graphs by various GSL algorithms.

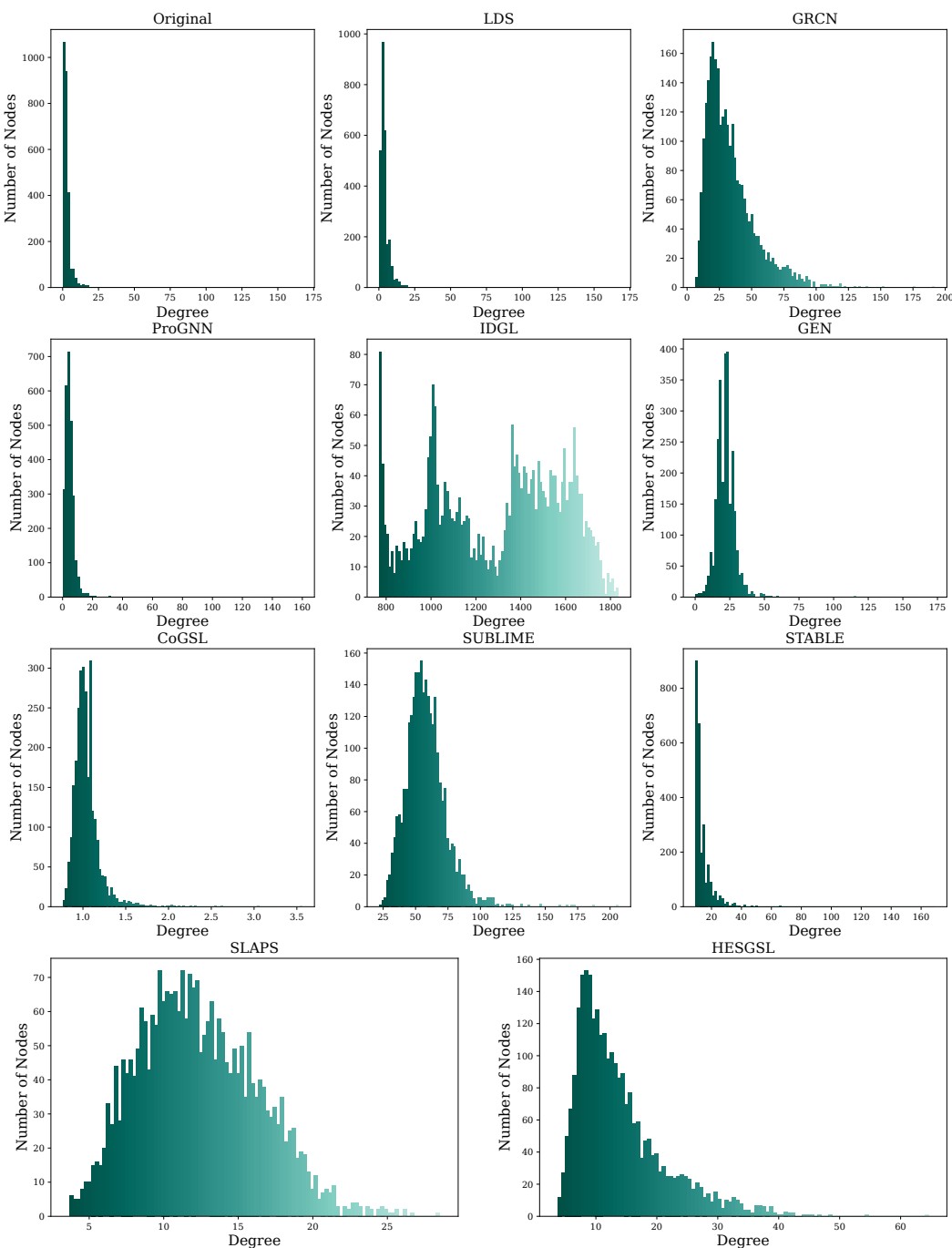

Figure 7: Degree distribution of the original graph of Cora and the learned graphs by various GSL algorithms.