# OpenReview forum: "GSLB: The Graph Structure Learning Benchmark"
_NeurIPS.cc/2023/Track/Datasets_and_Benchmarks — NeurIPS 2023 Datasets and Benchmarks Poster_

### Official Review · Reviewer_ee9D · 2023-07-03

**Rating:** 7
**Confidence:** 3
**Clarity:** The paper is generally well-written.

**Strengths:**

1. GSLB examines existing GSL methods with various tasks, including homogeneous and heterogeneous node classification and graph classification. The inclusion of heterogeneous node classification is a plus. In addition, the authors include a large dataset, ogbn-arxiv to evaluate the baselines, which is also a plus.
2. GSLB also examines the robustness of GSL methods in addition to standard node classification, which is also a plus.
3. The paper is well-organized and in general easy to follow.
4. The GSLB repository provides good documentation for readers to use, e.g. setting datasets, models, adversarial perturbations, etc.

**Additional Feedback:**

Please see my feedbacks above.

**Correctness:**

In my opinion, the authors have done the experiments in an appropriate and fair manner.

**Documentation:**

The authors provide sufficient documentation for readers to try the implemented algorithms on the provided datasets. It would be better if the authors can provide some instructions about how to implement new algorithms.

**Ethics:**

No.

**Opportunities For Improvement:**

In my opinion, this paper does not have major drawbacks. I list several minor points or opportunities for improvement.

1. It would be good if the authors can perform some additional analyses about the design choices mentioned in Table 1. For example, which type of structure modeling performs better, which regularizations are more useful than others, etc. Such an analysis would be helpful to practitioners who try GSL on their own problems.



**Relation To Prior Work:**

To my knowledge, this paper is among the first works on benchmarking graph structure learning.

**Summary And Contributions:**

In this paper, the authors present GSLB, a benchmark and analysis of graph structure learning (GSL) methods. The authors analyze a variety of GSL methods with 19 datasets and various tasks (node classification, heterogeneous node classification, graph classification). In addition, robustness of GSL methods have been studied with random and adversarial perturbations. Finally, scalability analysis is presented. The analyses show that GSL methods perform well on heterophilous graphs, are robust against perturbations, perform well especially with limited labels, and generally suffer from scalability issues.

---

> ### Author Response · Authors · 2023-08-11
> **Response to Reviewer ee9D**
>
> Thank you very much for your recognition of our work and the provided revision suggestions.
>
> ### Q: It would be good if the authors can perform some additional analyses about the design choices mentioned in Table 1. For example, which type of structure modeling performs better, which regularizations are more useful than others, etc. Such an analysis would be helpful to practitioners who try GSL on their own problems.
>
> A: Due to the diverse design principles of various algorithms in our benchmark, and the multitude of dataset, encoder, structural modeling, and regularization choices, conducting a module-level analysis is very challenging. Therefore, in this paper, we only performed an analysis at the algorithm level.
>
> However, researchers can still obtain insights for designing or applying GSL algorithms through our comparative study. For instance, we have found that self-supervised GSL algorithms exhibit excellent performance when facing topology attacks. Therefore, in the face of noisy or malicious graph structures, self-supervision can be used to assist GSL.  Besides, we find that edge-oriented GSL algorithms (e.g. ProGNN) have strong robustness with respect to feature robustness. And most GSL algorithms show surprising performance with few labels. All of these observations will help practitioners quickly deploy GSL on their own specific problems.

---

### Official Review · Reviewer_h3AR · 2023-07-19
**Review of the benchmark paper GSLB**

**Rating:** 6
**Confidence:** 3

**Strengths:**

Besides the contributions listed above, this paper is:
1. well-structured and coherent.
2. The code stored in GitHub is well-documented.
3. Authors have a good coding practice.

**Additional Feedback:**

N/A

**Clarity:**

The paper is not really well written. The authors should pay attention to the following issues:
1. Avoid mixing the use of abbreviations, e.g. GSL
2. Should unify UK/US English spelling
3. Fix grammatical mistakes: reflecting (line 35), appeals for (line 46), concerning with (line 50), analysis (line 58)
4. Fix typo: spread (line 43), does (line 52), learned (lines 657, 669), more (line 671)
5. Improve clarity: which (line 29), rare -> few (line 63), different level of (line 68), advances and basic concept (line 85), misuse of colons (line 96-98), mainstream -> popular (line 127)
6. "Angstroms" has no need to be capitalised in line 580.

**Correctness:**

In general, the evaluation methods and experiment design are appropriate and performed correctly. But more explanation are required as shown in "Opportunities For Improvement".

**Documentation:**

Documentation is available on GitHub. Each function is well-documented, and users can understand its functionality quickly.

**Ethics:**

No ethical concerns.

**Opportunities For Improvement:**

1. It is better to have a graph showing the correlation between model performances and dataset characteristics (heterophily, size, etc.).
2. Justification of the choice of metric (accuracy) and encoder (GCN) is needed.
3. Some of the datasets ('roman-empire', 'amazon-ratings', 'minesweeper', 'tolokers' and 'questions') are unreachable.
4. All hyperlinks are unreachable.
5. RQ5 is actually discussed in Appendix B, not in Section 4.4 (line 154).
6. More implementation details (setting of GAT_{knn}, LP etc.) need to be provided.
7. More elaborations is required to explain why GSL algorithms can outperform Heterogeneous Graph Attention Networks (lines 212-214), the term "perfect" is hard to understand.
8. More elaborations is required to explain the observations in Table 10. Which GSL algorithm is suitable for what downstream tasks?
9. Missing highlighting for the second-best results in Table 6.
10. Missing implementation code for DAGNN, self-training, GPRGNN, HAN, NodeFormer, and LDS methods in GitHub.

**Relation To Prior Work:**

Yes, in the introduction section it has a paragraph discussing pyGSL and it has a good collection of GSL methods.

**Summary And Contributions:**

This paper presents a pipeline for comprehensive benchmarking for Graph Structure Learning (GSL) in effectiveness, robustness and complexity. This benchmark summarised various state-of-the-art GSL methods, evaluated them from multiple perspectives and concluded some surprising phenomenons and difficulties faced in the GSL community.

The contributions of this paper include:
1. Providing a high-level abstraction of general GSL algorithms.
2. Summarising representative GSL methods according to their type of downstream task, method prior required, regularisation applied, method complexity, etc.
3. Exploring method sensitivity on the number of labels per class in node classification tasks.
4. Exploring method performances on graph-level classification task and homogeneous/heterogeneous graph node classification tasks.
5. Exploring the robustness of GSL methods when facing edge addition/deletion on given prior and graph topology attacks.
6. Summarising training time and memory requirements.
7. Summarising common challenges in GSL.
8. Providing a unified API for fair comparison of GSL methods.

---

> ### Author Response · Authors · 2023-08-11
> **Response to Reviewer h3AR**
>
> Thanks for your meticulous review and constructive suggestions!
>
> ### Q: It is better to have a graph showing the correlation between model performances and dataset characteristics (heterophily, size, etc.).
> A: We explore the relationship between model performance and homophily ratio of learned structure and found that they are not always positively correlated.
>
> Due to the page limitation, we only show the results on Cora datasets (in the TR scenario):
>
> | Cora        | CoGSL | ProGNN | Original(GCN) | STABLE | GEN   | GRCN  | SUBLIME | SLAPS | HES-GSL | IDGL  |
> | - | - | - | - | - | - | - | - | - | - | - |
> | Edge Hom.   | 0.918 | 0.817  | 0.810         | 0.799  | 0.744 | 0.727 | 0.705   | 0.691 | 0.630   | 0.246 |
> | Performance | 81.76 | 80.30  | 81.46         | 80.20  | 80.21 | 83.87 | 83.40   | 72.28 | 73.68   | 83.88 |
>
> These algorithms are arranged in descending order of homophily ratio. We can observe that performance and homophily are not positively correlated. We speculate this might be related to the inaccurate measurement of the homophily ratio in the current homophily evaluation methods. Please refer to Figure 7 (in the revised version) for more information.
>
> ### Q: Justification of the choice of metric (accuracy) and encoder (GCN) is needed.
> A: Since the majority of current node classification and graph classification works adopt accuracy as the evaluation metric, we also followed this evaluation scheme. The choice of GCN was made for the same reason. Additionally, in our GSLB, it is convenient to change different metrics and encoders.
>
> ### Q: Some of the datasets ('roman-empire', 'amazon-ratings', 'minesweeper', 'tolokers' and 'questions') are unreachable.
> A: We did not use these datasets for experiments, and they are not presented in the paper. We will remove the interfaces of these datasets from GitHub shortly.
>
> ### Q: All hyperlinks are unreachable.
> A: We did not encounter this issue while writing the paper using Overleaf. We suspect it might be related to the downloaded template. We have addressed this problem in the revised version. Thank you very much for your reminder.
>
> ### Q: RQ5 is actually discussed in Appendix B, not in Section 4.4 (line 154).
> A: We conduct graph-level experiments in Section 4.4, but due to page limitation, we have moved RQ5 to Appendix B. We have corrected this error in the revised version of the paper.
>
> ### Q: More implementation details (setting of GAT_{knn}, LP etc.) need to be provided.
> A: We briefly introduce the setting of kNN-input methods (Lines 161-162 of the revised version). Specifically, we use the kNN graph as input structure in the TI scenario, the selection of k was obtained through hyper-parameter tuning within the range {5, 10, 15, 20, 25, 30}. There is indeed a limited description of these methods in the paper. We list the baseline models used in our benchmark in Appendix A.2.
>
> ### Q: More elaboration is required to explain why GSL algorithms can outperform Heterogeneous Graph Attention Networks (lines 212-214), the term "perfect" is hard to understand.
> A: “not perfect either” in the paper indicates that the current heterogeneous graph dataset still contains noise or missing connections. From Table 4, we can find that: 1) because GTN and HGSL consider both heterogeneity and structure learning, they generally outperform other models on heterogeneous graph datasets; (2) GSL algorithms generally outperform the vanilla GNN models (e.g. GCN and GAT) since they have learned better structures to facilitate message passing; (3) Some datasets (e.g. Yelp) exhibit stronger heterogeneity, and on such datasets, models that consider heterogeneity (e.g. HAN, GTN, and HGSL) perform significantly better.
>
> ### Q: More elaboration is required to explain the observations in Table 10. Which GSL algorithm is suitable for what downstream tasks?
> A: Peptides-Struct aims to predict aggregated 3D properties of the peptides at the graph level, such as sphericity and Length. GSL may be more beneficial for predicting the structural properties of graphs. Therefore, GSL algorithms perform better on Peptides-Struct than on Peptides-Func.
>
> ### Q: Missing implementation code for DAGNN, self-training, GPRGNN, HAN, NodeFormer, and LDS methods in GitHub.
> A: We have integrated all of these models into GitHub. Please check out our GitHub repository, all of the baseline models are in GSLB/GSL/model/baselines.py.
>
> ### Q: Missing highlighting and writing issues
> A: We have fixed the issues of writing. Besides, we have used grammar correction and proofreading tools to recheck the paper. Thank you very much for your meticulous revisions and suggestions on the paper writing.

---

> > ### Comment · Reviewer_h3AR · 2023-08-22
> >
> > I have gone through the rebuttal, and the author's reply addresses my concerns.

---

### Official Review · Reviewer_KDj1 · 2023-07-20
**Review for Submission 505**

**Rating:** 6
**Confidence:** 4
**Correctness:** 1. Can you draw observation 1) from t…
**Clarity:** This paper is well written.

**Strengths:**

1. This paper proposes a standard setting of evaluation metrics for GSL.
2. This paper conducts extensive experiments, and offers insights and opportunities for further research. Additionally, it provides important evaluation indicators for subsequent research.
3. This paper evaluates models from multiple dimensions, such as effectiveness, robustness and complexity.

**Additional Feedback:**

None.

**Documentation:**

The important parameters in each algorithm are not reported in the paper. Therefore, it is hard to reproduce the experimental results.

**Ethics:**

None.

**Limitations:**

Yes.

**Opportunities For Improvement:**

1. This work covers various algorithms, but most of them are supervised methods. There is a lack of self-supervised GSL methods.
2. All evaluated GSL algorithms should be included in Table 4. The description of experimental results is too simple. For example, what is the relationship between model performance, efficiency, and space consumption?
3. GSL aims to learn a good graph topology, so the downstream tasks should include link prediction or clustering, in order to check the quality of the learned adjacency matrix.

**Relation To Prior Work:**

Compared with an existing GSL toolkit, called pyGSL, GSLB includes more models and datasets.

**Summary And Contributions:**

The authors conduct a comprehensive analysis of Graph Structure Learning (GSL) and provide a fair evaluation standard for subsequent experiments. Their contributions are as follows:
1. This work develops a library, Graph Structure Learning Benchmark (GSLB), consisting of 14 GSL algorithms and 20 diverse datasets.
2. This work systematically studies the representative GSL methods and discusses them extensively.

---

> ### Author Response · Authors · 2023-08-11
> **Response to Reviewer KDj1**
>
> Thanks for providing suggestions for improvement.
>
> ### Q: This work covers various algorithms, but most of them are supervised methods. There is a lack of self-supervised GSL methods.
>
> A: We actually integrate most self-supervised GSL methods. SLAPS utilizes more supervision signals for inferring a graph structure through self-supervised learning. STABLE optimizes the learned graph structure through contrastive learning. HES-GSL proposes homophily-enhanced self-supervision for GSL to provide more supervision information to topology inference. Please refer to Appendix A.2 for more details on these algorithms. Additionally, we also add one more self-supervised GSL algorithm, GSR, which first estimates the underlying graph structure by a multi-view contrastive learning framework, and then fine-tunes a GNN on the learned structure. The results of GSR are as follows:
>
> In the Topology Refinement (TR) scenario,
>
> | Cora           | Citeseer       | Pubmed         | ogbn-arxiv | Cornell        | Texas          | Wisconsin      | Actor          |
> | - | - | - | - | - | - | - | - |
> | 82.48$\pm$0.43 | 71.10$\pm$0.25 | 78.09$\pm$0.53 | OOM        | 44.32$\pm$2.16 | 60.81$\pm$4.87 | 56.86$\pm$1.24 | 30.23$\pm$0.38 |
>
> In the Topology Inference (TI) scenario,
>
> | Cora           | Citeseer       | Pubmed         | ogbn-arxiv | Cornell        | Texas          | Wisconsin      | Actor          |
> | - | - | - | - | - | - | - | - |
> | 66.28$\pm$0.59 | 66.77$\pm$0.62 | 68.49$\pm$1.49 | OOM        | 70.27$\pm$3.62 | 74.84$\pm$3.63 | 78.62$\pm$5.91 | 33.73$\pm$1.12 |
>
> ### Q: All evaluated GSL algorithms should be included in Table 4. The description of experimental results is too simple. For example, what is the relationship between model performance, efficiency, and space consumption?
> A: Thanks for your advice on experiments of heterogeneous node classification. We have refined the experiments in Section 4.3, and we evaluate all node-level GSL algorithms on heterogeneous graph datasets, the remaining results are as follows:
>
> |            | ACM              |                  | DBLP             |                  | Yelp             |                  |
> | - | - | - | - | - | - | - |
> |            | macro-f1         | micro-f1         | macro-f1         | micro-f1         | macro-f1         | micro-f1         |
> | STABLE     | 83.54 $\pm$ 4.20 | 83.38 $\pm$ 4.51 | 75.18 $\pm$ 1.95 | 76.42 $\pm$ 1.95 | 71.48 $\pm$ 4.71 | 76.62 $\pm$ 2.75 |
> | GEN        | 87.91 $\pm$ 2.78 | 87.88 $\pm$ 2.61 | 89.74 $\pm$ 0.69 | 90.65 $\pm$ 0.71 | 80.43 $\pm$ 3.78 | 82.68 $\pm$ 2.84 |
> | SUBLIME    | 92.42 $\pm$ 0.16 | 92.13 $\pm$ 0.37 | 90.98 $\pm$ 0.37 | 91.82 $\pm$ 0.27 | 79.68 $\pm$ 0.79 | 82.99 $\pm$ 0.82 |
> | NodeFormer | 91.33 $\pm$ 0.77 | 90.60 $\pm$ 0.95 | 79.54 $\pm$ 0.78 | 80.56 $\pm$ 0.62 | 91.69 $\pm$ 0.65 | 90.59 $\pm$ 1.21 |
>
> In addition, we actually analyzed the relationship between model performance, efficiency, and space consumption. In Figure 5 of Section 4.6 (in the revised version), we plot the relationship between model performance and complexity, where the horizontal coordinate is complexity and the vertical coordinate is performance. We can find that GRCN has achieved a good performance in both performance and efficiency and although LDS and IDGL have good performance in model performance, they have extremely high time and space complexity.
>
> ### Q: GSL aims to learn a good graph topology, so the downstream tasks should include link prediction or clustering, to check the quality of the learned adjacency matrix.
> A: Thank you for your question. We have considered this question, but it is not thoughtful. To better check the quality of the learned graph structures, we suggest to refer to Figure 6 (in the revised supplementary material). We have visualized the original graph of Cora and the learned graphs by various GSL algorithms. We can observe that GSL algorithms prefer to connect intra-class edges instead of inter-class edges, which indicates that GSL produces higher-quality structures than the original structure.
>
> ### Q: Can you draw observation 1) from the experimental results in Table 2 alone?
> A: We have added edge homophily ratio into Table 2. Please refer to our revised paper.
>
> ### Q: SUBLIME is a self-supervised method without label information. It is improper to compare it with other methods that require labels and draw relevant conclusions in Fig. 2.
> A: SUBLIME is only self-supervised during the structure learning phase, it trains a GCN on the learned graph structure in the pattern of supervised learning. Therefore, comparing SUBLIME with other models is fair (It is consistent with the experiments chapter in the original paper).
>
> ### Q: it is hard to reproduce the experimental results.
> A: We have released the code of all GSL algorithms and baseline models, and the hyper-parameters of these models can be found in the configs folder. Please refer to our GitHub repository.

---

> > ### Comment · Reviewer_KDj1 · 2023-08-28
> > **Review for Submission 505**
> >
> > Thanks for the responses.
> >
> > Could you provide the link to the revised paper? I cannot find the new content in https://openreview.net/pdf?id=xT3i5GS3zU.
> >
> > O1 & O2. The evaluated settings are not consistent. For example, Table 2 has STABLE, while Table 3 does not; Table 3 has HES-GSL while Table 2 does not; the evaluated datasets and methods in Table 4 are different from those in Table 2. The authors should explicitly explain these inconsistent experimental settings.
> >
> > O3. Why it is not possible to run link prediction tasks? Since link prediction is an important workload, especially for analyzing graph structures, it is better to include it in the evaluation.

---

> > > ### Author Response · Authors · 2023-08-28
> > > **Response to Reviewer KDj1**
> > >
> > > We sincerely appreciate your feedback and the time you've dedicated to reviewing our work.
> > >
> > > * **Could you provide the link to the revised paper? I cannot find the new content in https://openreview.net/pdf?id=xT3i5GS3zU.**
> > >
> > > We have updated our revised paper, and highlighted new content in https://openreview.net/pdf?id=xT3i5GS3zU (marked as red in the revised PDF). First, we integrated a new GSL algorithms, GSR, to increase the amount of self-supervised GSL algorithms. Second, we supplemented experimental results of STABLE in Table 3 and the remaining GSL algorithms in Appendix B.1. Third, we also added a robust analysis with respect to feature noise to further investigate the robustness of GSL.
> > >
> > > * **O1 & O2. The evaluated settings are not consistent. For example, Table 2 has STABLE, while Table 3 does not; Table 3 has HES-GSL while Table 2 does not; the evaluated datasets and methods in Table 4 are different from those in Table 2. The authors should explicitly explain these inconsistent experimental settings.**
> > >
> > > Thanks for your reminding. We have supplemented the results of STABLE in Table 3. Regarding the results of HES-GSL, because HES-GSL is focus on Topology Inference (TI) scenario, it aims to construct a graph structure for independent and identically distributed samples (e.g., text, image). And HES-GSL does not consider the scenario that the original graph structure is available. Therefore, we only present the results of HES-GSL in Table 3, which is same as the original paper [1].
> > >
> > > * **O3. Why it is not possible to run link prediction tasks? Since link prediction is an important workload, especially for analyzing graph structures, it is better to include it in the evaluation.**
> > >
> > > Actually there are few works do link prediction task to evaluate GSL algorithms. There are two main reasons: 1) In link prediction task, researchers usually randomly select some existing edges as positive samples, and randomly select some non-existing edges as negative samples. And they mask out the positive samples from the original graph. Due to the structure learning ability of GSL, under the guidance of positive and negative binary classification loss, GSL preferentially generates the edge of positive samples, which may cause the over-fitting. 2) Some GSL algorithms firstly pre-train the encoder, and then infer or refine the structure based on high-quality representations (such as STABLE and SUBLIME). However, most of GSL algorithms output weighted graph structures (as shown in Figure 6), conventional link prediction task cannot be performed on weighted graphs. The weights of positive and negative samples are different, so it is hard to simply use binary classification loss to train the model, which is beyond the scope of this work.
> > >
> > > But we are committed to continuously improve our benchmark in the future. For example, the generalization ability of different structures across different downstream tasks is an under-explored and interesting topic. We leave this direction for future work.
> > >
> > > Thank you for your careful review and constructive suggestions. If you have any further questions or need additional information, please don't hesitate to reach out.
> > >
> > > [1] Wu, Lirong, et al. "Homophily-Enhanced Self-Supervision for Graph Structure Learning: Insights and Directions." IEEE Transactions on Neural Networks and Learning Systems (2023).

---

> ### Author Response · Authors · 2023-08-26
> **Thank you & looking forward to reply**
>
> Dear Reviewer KDj1:
>
> Thank you very much for your precious time and valuable comments. According to your reviews, we have taken the effort to provide detailed explanations and new experimental results, aiming to address your concerns comprehensively. Please let us know if you still have any unclear points in our work. We are happy to discuss further.
>
> Best,
>
> Authors

---

### Official Review · Reviewer_QMnH · 2023-07-22

**Rating:** 6
**Confidence:** 3
**Correctness:** good.
**Clarity:** good.

**Strengths:**

The comparisons between different methods are clear and easy to understand.

The authors conduct experiments on various settings including standard node classification, heterogeneous node classification, and graph classification.

The authors conduct training time and space analyses for different methods.


**Additional Feedback:**

NA

**Documentation:**

good.

**Limitations:**

see above.

**Opportunities For Improvement:**

I wonder about the performance of these methods on larger datasets such as ogbn-products. All these datasets used to evaluate the methods are relatively small, even ogbn-arxiv is still considered as a small dataset for ogb.

Most of the experiments (e.g., robustness analyses, time and space analyses) are based on very small toy datasets such as cora or citeseer. More experiments on larger datasets are required for a solid analysis.

**Relation To Prior Work:**

good.

**Summary And Contributions:**

This paper studies the representative GSL methods and summarizes them. This paper presents a library, Graph Structure Learning Benchmark (GSLB), consisting of 14 GSL algorithms and 20 diverse datasets.

---

> ### Author Response · Authors · 2023-08-11
> **Response to Reviewer QMnH**
>
> ### Q: I wonder about the performance of these methods on larger datasets such as ogbn-products. All these datasets used to evaluate the methods are relatively small, even ogbn-arxiv is still considered as a small dataset for ogb.
>
> A: Thanks for your question about the scalability of GSL.
>
> As we have mentioned in Section 4.6, because of the $O(N^2)$ complexity of most GSL algorithms, applying them to large-scale graphs becomes challenging. There are only a few GSL algorithms are capable of running on the ogbn-arxiv dataset (SUBLIME, NodeFormer, SLAPS, and HES-GSL), so when conducting experiments on larger graph datasets, only a very small number of algorithms can run, limiting the meaningfulness of the analysis.
>
> We conduct a scalability analysis in Section 4.6, and we can observe that edge-oriented algorithms (e.g. LDS and ProGNN) cost high memory requirements. Therefore, we believe that directly optimizing graph structures is challenging to apply to large-scale graph datasets. We also mentioned in Section 5 that future works should focus on overcoming the limitations of GSL in terms of complexity.

---

> ### Author Response · Authors · 2023-08-22
> **Thank you & looking forward to reply**
>
> Dear Reviewer QMnH:
>
> Thank you very much for your precious time and valuable comments. According to your reviews, we responded with detailed explanations, which we believe have covered your concerns. We kindly remind you that the deadline for discussions is approaching. Please let us know if you still have any unclear points in our work. We are happy to discuss further.
>
> Best,
>
> Authors

---

### Official Review · Reviewer_wra6 · 2023-07-22
**Comprehensive benchmarking paper for graph structure learning**

**Rating:** 7
**Confidence:** 4
**Clarity:** The paper is well written, and contri…

**Strengths:**

- Comprehensive benchmarking study, covering all major aspects: performance, time/space requirement and structure refinement/inference capability.
- It’s a great idea to study the structure refinement capability using a perturbation-based approach.
- The paper uses a mostly divers set of datasets.
- The paper makes an effort to quantitatively and intuitively analyze the properties of the learned graph structures comparing, for example, densities and partial adjacency matrices.


**Additional Feedback:**

Scalability to larger graphs seems to be the major challenge in GSL. It is a good to use the large arXiv dataset where most GSL models fail OOM, because it guides future work towards addressing this challenge. It could be interesting to add short discussion on how, some methods are able to scale to that size.

[1] Dwivedi, Vijay Prakash, Chaitanya K. Joshi, Anh Tuan Luu, Thomas Laurent, Yoshua Bengio, and Xavier Bresson. "Benchmarking Graph Neural Networks." JMLR (2023).

[2] Hu, Weihua, Matthias Fey, Marinka Zitnik, Yuxiao Dong, Hongyu Ren, Bowen Liu, Michele Catasta, and Jure Leskovec. "Open graph benchmark: Datasets for machine learning on graphs." NeurIPS (2020).

[3] Liu, Renming, Semih Cantürk, Frederik Wenkel, Sarah McGuire, Xinyi Wang, Anna Little, Leslie O’Bray et al. "Taxonomy of benchmarks in graph representation learning." LoG (2022).

**Correctness:**

The contribution is correct and the empirical claims are sufficiently supported by experimental evidence.

**Documentation:**

The paper is overall well documented. One could maybe add avg. degree to Table 8?

**Limitations:**

- The paper offers limited analysis on how to differentiate between the various GSL methodologies.
- As pointed out by the authors, GSL approaches remain difficult to scale to larger datasets that are used for benchmarking today’s GNNs.


**Opportunities For Improvement:**

- It could benefit the paper to first develop a deeper understanding of the properties of the datasets, as they play a particularly important role for benchmarking models that learn the graph structure. E.g., [3] also used structural perturbations (in the context of GNNs) and found that Cora, PubMed and CiteSeer mostly rely on informative node features, with minor performance loss when perturbing or completely disregarding structure. Such insights could be valuable for understanding why GSL models perform well here, e.g., using node features to infer a “better” graph.
- The used datasets are reasonably divers but redundant in parts. Toy datasets Cornell, Texas and Wisconsin do not give any complementary insights.
- GSL methods rely on very diverse methodologies, but the paper offers limited insights towards differentiating them and why some work better or worse in specific situations. For example, Figure 5 and Table 11 would be a good foundation for such analysis.


**Relation To Prior Work:**

- As GSL methods are compared to GNNs in this work, common benchmarking frameworks should be discussed in detail [1, 2]. It would also be good to explain why GSL needs a different mix of datasets.
- The robustness analysis with respect to structure perturbations is also very similar to [3], where such techniques are used taxonomize graph learning datasets in the context of GNNs.


**Summary And Contributions:**

- Open-source library including 14 models and 20 commonly used graph datasets covering node- and graph-level tasks.
- Models are benchmarked with respect to downstream performance, time & space requirement, and robustness to structural perturbations.
- The authors find that GSL can benefit from reducing the label rate and that some models can overcome structural perturbations.

---

> ### Author Response · Authors · 2023-08-11
> **Response to Reviewer wra6**
>
> Thanks for your constructive suggestions.
>
> ### Q: It could benefit the paper to first develop a deeper understanding of the properties of the datasets.
> A: Based on exploring structural robustness, we have added experiments to investigate feature robustness. We randomly mask a certain proportion of node features by filling them with zeros, to investigate the performance of GSL algorithms when node features are subjected to varying degrees of damage. Because of the page limitation, we only list the results of a few representative algorithms. The experimental results are as follows:
>
> On the Cora dataset (in the TR scenario)
>
> | Method  | 0.0   | 0.1   | 0.2   | 0.3   | 0.4   | 0.5   | 0.6   | 0.7   | 0.8   | 0.9   |
> | ------- | ----- | ----- | ----- | ----- | ----- | ----- | ----- | ----- | ----- | ----- |
> | GCN     | 80.86 | 80.23 | 78.42 | 77.71 | 76.98 | 74.02 | 74.58 | 71.48 | 66.21 | 61.61 |
> | GRCN    | 83.76 | 82.78 | 83.42 | 82.18 | 81.01 | 78.62 | 79.50 | 77.02 | 70.38 | 52.35 |
> | IDGL    | 83.37 | 82.27 | 81.71 | 82.35 | 78.43 | 76.40 | 76.31 | 72.91 | 66.57 | 59.38 |
> | ProGNN  | 80.30 | 79.88 | 79.52 | 79.59 | 79.29 | 77.80 | 78.54 | 76.00 | 76.61 | 68.95 |
> | SUBLIME | 83.08 | 81.79 | 82.29 | 81.55 | 80.26 | 80.22 | 78.11 | 77.27 | 73.14 | 62.90 |
>
> On the Cora dataset (in the TI scenario)
>
> | Method  | 0.0   | 0.1   | 0.2   | 0.3   | 0.4   |
> | ------- | ----- | ----- | ----- | ----- | ----- |
> | MLP     | 58.01 | 54.71 | 49.71 | 46.98 | 42.05 |
> | GCN_knn | 65.79 | 62.10 | 58.08 | 50.78 | 46.57 |
> | SLAPS   | 72.35 | 69.70 | 65.52 | 63.30 | 58.48 |
> | SUBLIME | 72.74 | 68.96 | 66.14 | 62.43 | 58.75 |
> | HES-GSL | 73.97 | 72.59 | 69.44 | 63.99 | 57.85 |
>
> The results are similar to the other datasets. We can observe that: 1) the noisy node features play a more important role than the noisy structure (please refer to Figure 3); 2) interestingly, while most existing GSL methods rely on feature similarity between pairs of nodes to learn graph structure, they still exhibit good robustness when faced with noisy node features; 3) edge-oriented algorithms (e.g. ProGNN) show strong feature robustness, because they optimize adjacency matrix directly, and have less dependence on pairs of node features. Please refer to Section 4.5 in the revised version for more information.
>
> ### Q: The used datasets are reasonably divers but redundant in parts.
> A: Regarding your opinion that "the used datasets are redundant in parts", actually Cornell, Texas, and Wisconsin are the benchmark datasets of heterophilic GNN research. We have exhibited the homophily ratio (we use edge homophily here) of each homophilic and heterophilic graph dataset in Table 7. We can observe that Cornell, Texas, and Wisconsin have lower homophily ratios compared to hemophilic datasets like Cora, Citeseer, and Pubmed. Therefore, as shown in Table 2, we can find that most GSL algorithms can enhance the performance on heterophilic graph datasets. It proves the board applicability of GSL.
>
> ### Q: GSL methods rely on very diverse methodologies, but the paper offers limited insights towards differentiating them and why some work better or worse in specific situations.
> A: That's a good question, but it's not easy to investigate. In tasks such as node classification, the experimental results may not necessarily reflect the design advantage. However, we have indeed observed variations in performance about different types of algorithms on certain tasks. For example, in robustness analysis, we have found that self-supervised GSL algorithms perform exceptionally well, we attribute this phenomenon to self-supervised methods to avoid relying on unreliable supervision signals. Based on our experimental results, we have found that edge homophily is not the key factor of performance enhancement. As shown in Table 11, all learned graph structures by GSL algorithms have lower homophily ratios compared to the original graphs, regardless of node, edge, or EI homophily ratios. But we find that all of them have higher density compared with original graphs, we suspect that in the semi-supervised setting, denser graph structures may facilitate the propagation of supervision signals, thus leading to performance improvement.
>
> ### Q: As GSL methods are compared to GNNs in this work, common benchmarking frameworks should be discussed in detail [1, 2]. It would also be good to explain why GSL needs a different mix of datasets.
> A: We have added detailed information about baseline models in Appendix A.2. Due to the uncertainty and complexity of data collection, graph structures have several drawbacks, such as redundant, biased, noisy, and missing connections. Thus we evaluate GSL in different application scenarios with specific graph datasets.
>
> ### Q: One could maybe add avg. degree to Table 8?
> A: we have added avg. degree to Table 8:
>
> |              | ACM  | DBLP | Yelp |
> | ------------ | ---- | ---- | ---- |
> | Avg. #Degree | 1.51 | 1.44 | 3.72 |

---

### Decision · Program_Chairs · 2023-09-22

**Decision:**

Accept (Poster)

**Comment:**

The paper presents a comparative study of representative graph structure learning methods, benchmarking their performance on various downstream tasks, including both node- and graph-level objectives. The comparison is conducted along three dimensions: effectiveness, robustness, and complexity. This work has received very positive feedback from all reviewers, acknowledging its significant contributions